

# Gap-Free Global Annual Soil Moisture: 15km Grids for 1991-2016

Mario Guevara[1], Michela Taufer[2], Rodrigo Vargas[1]

[1]Department of Plant and Soil Sciences, University of Delaware, Newark. United States.
[2]Department of Electrical Engineering and Computer Science, The University of Tennessee, Knoxville. United States.

*Correspondence to*: Rodrigo Vargas (rvargas@udel.edu)

**Abstract.** Soil moisture is key for quantifying soil-atmosphere interactions and the ESA-CCI (European Space Agency Climate Change Initiative) provides historical (>30 years) satellite soil moisture gridded data at the global scale. We evaluate an alternative approach to increase the spatial resolution of the original ESA-CCI soil moisture measurements from 27km to 15km grids by coupling machine learning (ML) with information from digital terrain analysis at the global scale. We modeled mean annual ESA-CCI soil moisture values across 26 years of available data (1991-2016) using a ML kernel method and multiple terrain parameters (e.g., slope, wetness index) as prediction factors. We used ground information from the International Soil Moisture Network (ISMN, n=13376) for evaluating soil moisture predictions. We provide gap-free mean annual soil moisture predictions, which increase by nearly 50% the spatial resolution of ESA-CCI soil moisture product. Our predictions showed a statistical accuracy varying 0.69-0.87% and 0.04 $m^3/m^3$ of cross-validated explained variance and root mean squared error (RMSE). We found no significant differences between the ESA-CCI and our predictions, but we found discrepancy between multiple evaluation metrics (e.g., bias vs efficiency) comparing the ESA-CCI with the ISMN. We found a negative bias (-0.01 to -0.08 $m^3/m^3$) between the values of ISMN when comparing with the ESA-CCI and our predictions across the analyzed years. A temporal analysis, using a robust trend detection strategy (i.e., Theil-Sen estimator), suggests a decline of soil moisture at the global scale that is consistent in both gridded datasets and field measurements of soil moisture varying from -0.7[-0.77, -0.62]% in the ESA-CCI product, -0.9[-1.01, -0.8]% in the downscaled predictions, and -1.6 [-1.7, -1.5]% in the ISMN. The soil moisture predictions provided here (Guevara, et al., 2019, https://doi.org/10.4211/hs.b940b704429244a99f902ff7cb30a31f) could be useful for quantifying soil moisture spatial and temporal dynamics across areas with low availability of soil moisture information in the original ESA-CCI soil moisture current and future versions.



## 1 Introduction

Assessing the reliability of currently available soil moisture datasets is fundamental for a comprehensive understanding of

the global water cycle (Al-Yaari et al., 2019). Soil moisture datasets are useful to characterize hydrological patterns (Greve and Seneviratne, 2015), the influence of soil moisture on terrestrial carbon dynamics (van der Molen et al., 2011), and global climate variability (Seneviratne et al., 2013). Soil moisture information is used for identifying trends in the water cycle and could be useful to better characterize the response of ecosystems productivity to soil moisture decline (Zhou et al., 2014). However, quantifying the response of ecosystems productivity to soil moisture decline is challenging due to a current lack of

accurate and detailed long-term soil moisture datasets across large areas of the world. This lack of soil moisture information can affect our capacity to detect regional-to-global soil moisture trends and could be an important source of uncertainty in global models of land-atmosphere interactions (May et al., 2016).

Large-scale hydrological and ecological analyses (e.g., continental, global) and syntheses of global climate variability benefit

from soil moisture information provided by multiple soil moisture monitoring networks (Dorigo et al., 2011a) and from satellite soil moisture measurements (Dorigo et al., 2017; Liu et al., 2011). Field soil moisture measurements (from soil moisture monitoring networks; Fig. 1) and satellite soil moisture measurements are two main sources of continuous soil moisture information at the global scale used for quantifying regional-to-continental soil moisture patterns and trends.

Field soil moisture measurements are representative of small footprints within specific study sites, at specific soil depths (e.g., 0-5 cm) and the availability of field soil moisture measurements is sparse and limited across large areas of the world. On the other hand, satellite microwave radiometry using L-band (~ 1.4-1.427 GHz) and C-band (~4-8 GHz) for example, is optimal to estimate regional soil moisture (Mohanty et al., 2017). The number of efforts for providing satellite soil moisture data at the global scale (based on microwave radiometry) have increased during the last decade (Al-Yaari et al., 2019).

Satellite soil moisture datasets are representative for the first few cm of soil depth (e.g., 0-5 cm) and they are provided in grids with spatial resolution varying between 9 and 25 km (Senanayake et al., 2019). This is a range of spatial resolution commonly used in global studies quantifying land and atmosphere interactions (Crow et al., 2012; Jung et al., 2010). However, there are also large areas of the world (across specific environmental conditions) where no satellite soil moisture data are available due to intrinsic sensor limitations (McColl et al., 2017). Collection of field soil moisture information

across these missing areas is expensive, time consuming, and in many cases impossible due to logistical reasons. Consequently, many information gaps exist and modeling/validation efforts for predicting soil moisture values across unmeasured areas are required to increase the applicability of soil moisture datasets in studies at multiple scales (Singh et al., 2015).



Currently, the historical soil moisture product from the European Space Agency-Climate Change Initiative (ESA-CCI) provides soil moisture grids at the spatial resolution of ~27x27 km grids (Liu et al., 2011). The ESA-CCI soil moisture product is a synthesis from multiple soil moisture sources and it covers four decades (from the 1978 to 2018) of accurate soil moisture values at the global scale. The ESA-CCI soil moisture product contains uncertainty of measurements and it covers a longer period of time (from 1978 to 2018) compared with other satellite-derived soil moisture products (Al-Yaari et al.,

2019), and is suitable for applications in long-term ecological and hydrological studies (Dorigo et al., 2017).

Across large areas of the world, the ESA-CCI soil moisture product is being validated and calibrated against ground truth (i.e., field) soil moisture measurements (Al-Yaari et al., 2019; Dorigo et al., 2011a).  Previous work has included a quality control framework to remove potentially wrong measurements from the ESA-CCI soil moisture product, with the ultimate

goal of improving its spatial representativeness and reliability (Gruber et al., 2017). The ESA-CCI soil moisture product is constantly being improved in each released version and we highlight that versions 4.4 and 4.5 have substantial spatial gaps across the world (Fig. S1). Therefore, the development of alternative modeling and validation frameworks is needed to provide new datasets and information to complement the different versions of the ESA-CCI soil moisture product.

To improve the spatial resolution of satellite soil moisture gridded datasets, multiple efforts have used statistical learning methods to couple coarse satellite soil moisture datasets with multiple sources of environmental information at higher spatial resolutions. These sources of environmental information include vegetation indexes (from optical imagery) and climate information (Alemohammad et al., 2018). Chloropeth maps (i.e., land use, land forms) and soil information have also been used as prediction factors to improve the spatial resolution of soil moisture gridded datasets under statistical modeling (e.g.,

regression) frameworks (Peng et al., 2017). The sources of environmental information are the basis for downscaling satellite soil moisture grids because they are available at higher spatial resolution than the original satellite soil moisture grids. For example, optical remote sensing (i.e., light detecting and ranging) is able to provide vegetation indexes with a spatial resolution of meters (Dubayah and Drake, 2000), as well as elevation data is now available at higher levels of spatial resolution (e.g., meters; Tadono et al., 2014). For this study, we focus on the potential of using elevation data to represent

topographic variability and consequently represent the role of topography in the overall distribution of water across the landscape (Moeslund et al., 2013, Mason et al., 2016, Guevara and Vargas 2019).

Elevation data is the basis of digital terrain analysis to quantify topographic variability and land surface characteristics (e.g., terrain roughness, terrain slope, terrain convexity; Wilson, 2012). Previous studies have found evidence of a topographic

signal in satellite soil moisture measurements (Mason et al., 2016). Other studies have highlighted the potential of using these land surface characteristics as prediction factors for soil moisture and developing soil moisture products with higher spatial resolution compared with the original satellite soil moisture measurements (Guevara and Vargas, 2019; Western et al., 2002). Digital terrain analysis are calculations of land surface characteristics that largely depend on topography, such as



the terrain slope and aspect, or the topographic wetness index, which is a parameter that characterizes areas where soil

moisture could increase by the effect of the overland flow accumulation. The overland flow as well as the potential incoming solar radiation are two important topographic drivers of the spatial distribution of soil moisture (Nicolai-Shaw et al., 2015), its memory after precipitation events (McColl et al., 2017), and its role as a dominant control of plant productivity (Forkel et al., 2015).

Our overarching goal is to test how the application of digital terrain parameters (e.g., terrain slope and aspect, or the topographic wetness index) as predictors of soil moisture information could increase the spatial resolution of global mean annual satellite soil moisture information. The specific objectives are: 1) improve the spatial resolution (from 27 to 15km grids; an improvement of about 50%) of the ESA-CCI soil moisture product (version 4.2 at the annual scale; years 1991-2016); 2) test how this downscaled mean annual global estimate compares with the statistical distribution of the ESA-CCI

soil moisture measurements; and 3) test the consistency of temporal analyses at the global scale using our downscaled product, the ESA-CCI product, and field soil moisture measurements around the world. In this study, we provide a new dataset of gap-free mean annual global soil moisture estimates at 15km resolution for the years 1991-2016 (Guevara, et al., 2019,  https://doi.org/10.4211/hs.b940b704429244a99f902ff7cb30a31f)

**2 Methods**

We used a data-driven modelling approach including Geographical Information Systems (GIS) and platforms for geoscientific analysis and statistical computing for combining digital terrain parameters, soil moisture gridded estimates and soil moisture tabular datasets. Field information of soil moisture is used for validating our soil moisture predictions based on one data driven

model for each year of available satellite soil moisture data in the ESA-CCI.

**2.1 Datasets**

This study is based on the analysis of ESA-CCI soil moisture measurements (version 4.2, years 1991-2016; Fig. 2). We

postulate that mean annual soil moisture (for any specific year) can be predicted using a regression model and digital terrain parameters (derived from elevation data) as prediction factors. In this regression framework, soil moisture is represented by the yearly mean of soil moisture values (for each calendar year) at the central coordinates (latitude and longitude) of pixels from the ESA-CCI soil moisture measurements. A similar approach was applied at the regional-scale within the conterminous United States providing an improvement in spatial resolution and ground truth validation of satellite soil

moisture derived from the ESA-CCI (Guevara and Vargas 2019).
The explanatory variables for soil moisture were represented with the values of the terrain parameters for the locations of the aforementioned central coordinates of the original ESA-CCI soil moisture measurements. Therefore, soil moisture for each



model/year can be predicted at finer spatial resolutions defined by the spatial resolution of the digital terrain parameters (Guevara and Vargas, 2019). We first harmonized (e.g., same geo-spatial reference, same extent) the ESA-CCI soil moisture
and topographic data (elevation and derived terrain parameters) using open source geographic information systems (Hijmans, 2019). The digital terrain parameters (Fig. 3) were derived from elevation data in SAGA-GIS (System for Automated Geoscientific Analysis-GIS) (Conrad et al., 2015). The source of elevation data was a radar based digital elevation model (Becker et al., 2009) that we resampled to a spatial resolution of 15 km across the world. We recognize that this resolution is still too coarse to represent the local variability of soil moisture but this dataset has two advantages: 1) it is
nearly a 50% improvement of spatial resolution when compared with the original ESA-CCI (~27 km grids) soil moisture product, and 2) our framework produced a gap-free global annual soil moisture estimate at 15 km resolution and it is theoretically applicable for predictions at higher temporal resolution (e.g., of months, weeks or days).

**2.2 Refinement modeling**


To predict soil moisture at a finer spatial resolution (15 km) than the original ESA-CCI product (27 km), we used a machine learning (ML) kernel-based method known as weighted Kernel Nearest Neighbors (KKNN; Hechenbichler and Schliep, 2004). This ML method was used to account for likely non-linear relationships between soil moisture and terrain parameters at the global scale. The KKNN method is a time efficient algorithm compared with more complex ML algorithms (i.e., tree-
based, deep learning based). It is a pattern recognition technique based on multiple data neighbors to account also for variations in the relationship of soil moisture with its explanatory variables from one place to another. The KKNN algorithm has two main user defined parameters, the parameter $k$ determines the number of neighbors from which information will be considered for prediction. The second parameter is a kernel function (e.g., triangular, epanechnikov or Gaussian among others) that allows to convert distances into weights (the farther the neighbor, the smaller the weight it will be assigned) that
can then be used to take a weighted vote or a weighted average respectively for classification (i.e., predicting categorical variables such as soil type) of for regression problems (i.e., predicting continuous variables such as soil moisture).

**2.3 Model parameter selection**

For each yearly mean, each model was first parameterized (selection of optimal parameters for each model/year) using cross validation and folds of 10 % of available data out of each iteration. The cross-validation indicators (information criteria about model performance) were the Pearson correlation coefficient (r) and the root mean squared error (RMSE) for each modeled yearly mean. Using the combination of $k$ and kernel function of the model generating the lowest RMSE and highest r for each model/year, we predicted soil moisture at the resolution of the aforementioned terrain parameters (i.e., 15 km) at
the global scale.




Then, the resulting soil moisture predictions and the ESA-CCI soil moisture product were validated against the available soil moisture data reported in the International Soil Moisture Network (ISMN; Dorigo et al., 2011a, 2017) for each annual mean (1991-2016). We extracted the values of soil moisture gridded measurements to the locations on the ISMN available data. A total of 8080 tables with soil moisture information with multiple data sizes (from multiple contributing networks) were extracted from the ISMN for the analyzed period of time (1991-2016). We also provided this information harmonized in an annual basis, including the values of the ESA-CCI and our predictions for the locations of the ISMN dataset (see section 5).

## 2.4 Assessment metrics

The ISMN data was used to calculate multiple model evaluation indicators (see Carslaw ,2015) for comparing the ESA-CCI soil moisture product and the soil moisture predictions based on digital terrain analysis approach. These evaluation statistics were performed using the *openair* package of the R software (Carslaw and Ropkins, 2012). These evaluation statistics included the number of complete pairs of data (n) and the Pearson correlation coefficient (r) between predicted and observed values as well as systematic error indicators:

- MB, the mean bias;
- MGE, the mean gross error;
- NMB, the normalized mean bias;
- NMGE, the normalized mean gross error;
- RMSE, the root mean squared error;

The fraction of predictions within a factor of two of the observed values (FAC2) was another evaluation indicator included in our analysis. The FAC2 is a robust metric for model evaluation because it is not overly influenced by outliers (Chang and Hanna, 2004). Other evaluation indicators that are not sensitive to outliers and extreme values were also included:

- The Coefficient of Efficiency (COE, based on Legates and McCabe, (1999) and Legates and McCabe (2013)). The COE has been used widely used to evaluate the performance of hydrological models. A perfect model has a COE = 1. A value of COE= 0 or negative implies low prediction capacity;
- IOA, the Index of Agreement (Willmott, Robeson, and Matsuura 2012), which spans between -1 and 1 and with values approaching +1 representing better model performance. The IOA indicates the proportion of the sum of the error magnitudes in relation to the sum of the observed-deviation magnitudes.



By interpreting the aforementioned model evaluation indicators, a perfect model would have a FAC2, r, COE and IOA ~ 1.0,
while all the others ~ 0. These metrics represent a valuable set of information criteria for comparing both the ESA-CCI soil
moisture product and the soil moisture predictions based on digital terrain analysis against ground data from the ISMN.

**2.4 Trend detection**

We also performed a non-parametric trend detection test (i.e., Theil-Sen estimator) to compare soil moisture trends between
the ESA-CCI soil moisture product and the downscaled soil moisture predictions based on digital terrain analysis (yearly
means 1991-2016). The same trend detection test was applied to the field information contained in the ISMN dataset for
comparative purposes. A pixel-wise trend detection test was also applied to search for changes in the regression relationship
(i.e., soil moisture ~ years) using different regression parameters before and after any possible breakpoint. A minimum of
four years are required between break points for detecting trends and segments between break points with less than eight
observations are not considered. Therefore, this method is considered to provide a robust trend detection estimate (Forkel et
al., 2013, 2015).

**3 Results**

We provided a dataset of gap-free downscaled mean annual soil moisture predictions at the global scale at 15 km grids for
years 1991-2016. These predictions are based on ML and digital terrain parameters and they are provided in a generic raster
format (Guevara, et al., 2019, https://doi.org/10.4211/hs.b940b704429244a99f902ff7cb30a31f). Supporting the reliability of
our prediction framework, the original ESA-CCI and our downscaled soil moisture predictions based on digital terrain
analysis showed consistently a global mean annual value of 0.19 $m^3/m^3$ and a standard deviation of 0.6 $m^3/m^3$ considering
the mean value of all analyzed years. The downscaled soil moisture predictions showed a similar bimodal distribution
compared with the original ESA-CCI soil moisture product (Fig. 4). The ISMN showed also a bimodal distribution but a
larger range (>50%) of soil moisture values compared with both soil moisture gridded measurements (Fig. 4). The ISMN
values (in an annual basis) showed a mean value of 0.24 and a standard deviation 0.12 $m^3/m^3$. Thus, the complete range of
soil moisture variability in the ISMN is higher than the ESA-CCI satellite measurements and therefore is also higher than the
downscaled soil moisture predictions based on digital terrain analysis (Fig. 4).

**3.1 Model parameter selection**

The cross-validated Pearson correlation coefficient (r) between observed and predicted soil moisture across the analyzed
years varied from 0.78 to 0.81, demonstrating a reliable prediction capacity. The RMSE varied from 0.03 to 0.04 $m^3/m^3$, in
all cases below the first quartile of the ESA-CCI soil moisture training data distribution (0.15 $m^3/m^3$, Table 1).





## 3.2 Evaluation against field data

The validation of the ESA-CCI soil moisture product and the ISMN (Table 2) showed relatively similar results compared to the validation of the downscaled soil moisture predictions based on digital terrain analysis and the ISMN (Table 3). In Table 2 and Table 3 the calculated evaluation statistics (n, FAC2, MB, MGE, NMB, RMSE, r, COE, and IOA) are provided.

Table 2 and Table 3 provide model evaluation statistics (agreement metrics between observed and modeled) for 26 different
yearly models for the years 1991 to 2016. The number of complete pairs of data available for validation (n) increases with time. (i.e., respectively for Table 2 and Table 3, we counted in 102 and 104 available pairs of points for 1991 while for 2016 these numbers increased to 1165 and 1194 pairs of points). The higher data density (e.g., with >1000 spatial coordinates with soil moisture from the ISMN and pixel soil moisture values from the ESA-CCI) was found between 2011 and 2016 (Table 2). In all cases, the evaluation statistics are equal or better for the downscaled soil moisture predictions based on digital
terrain analysis (Table 3) than the original ESA-CCI soil moisture product (Table 2).

Evaluation statistics such as FAC2, RMSE, r, COE and IOA showed variability across the analyzed years (Table 2 and Table 3). The FAC2 from 1991 to 2016 never falls below 0.70. This implies that no less than 70% of the predicted values were within a factor of 2 of the observed values. For the ESA-CCI soil moisture product, the lowest FAC2 value was 0.70 for
2012 and the highest value was 0.93 for 2000. These numbers were consistent for the downscaled predictions based on digital terrain analysis (0.79 and 0.94 respectively for the same years).  From both Table 2 and Table 3 the RMSE varied from 0.09 to 0.13 m$^3$/m$^3$. We found in all cases a negative mean bias, confirming that the ESA-CCI soil moisture product (and consequently our predictions) tend to underestimate the values of yearly soil moisture means from field measurements in the ISMN. The values of the mean bias varied from -0.01 to -0.1. Lowest bias was found for 2013 and highest bias for
1996. For the models after 1997, the r values range from 0.30 to 0.60. These values indicate a weak to moderate positive relationship between soil moisture gridded measurements and field soil moisture data. Before 1998, values on both tables indicate weak to no linear relationship. In all cases, we also found that the COE was less than 0 from 1991 to 2001. This suggests that the simple observed mean has a better agreement than the ESA-CCI with the values of the ISMN. For years 2001 onward, the models showed a relatively higher COE (Table 3). For the IOA values in both tables, all of the 26 values
are above 0, and range from 0.10 to 0.57. Therefore, this indicates a low to moderate, positive correlation between gridded soil moisture measurements and field soil moisture data. From this evaluation section, our results show that the original ESA-CCI soil moisture product tends to underestimate the values of the ISMN. The downscaled predictions based on digital terrain analysis are not significantly different compared with the ESA-CCI soil moisture product when evaluated against observed field values from the ISMN, but they provide 1) gap free soil moisture-related information and 2) higher spatial
resolution (from 27 to 15 km grids).



### 3.3 Trend detection results

At the locations with available field soil moisture information from the ISMN, we found a consistent soil moisture decline. This result was consistent when considering the values of the same locations for both tested gridded soil moisture datasets (i.e., ESA-CCI and the downscaled soil moisture predictions based on digital terrain analysis predictions). Overall, the trend

detection test suggests a consistent decline of soil moisture at the global scale (Fig. 6). The confidence intervals of the detected trends overlap between both gridded datasets, but both gridded datasets significantly underestimate the soil moisture decline detected from the field soil moisture measurements contained in the ISMN. These detected trends (using yearly means between 1991-2016) decreased from -1.6 [-1.7, -1.5]% in the field measurements of the ISMN (Fig. 6a), to -0.9[-1.01, -0.8]% in the downscaled soil moisture predictions based on digital terrain analysis (Fig. 6b), to -0.7[-0.77, -0.62]% in the

ESA-CCI soil moisture product (Fig. 6c). Thus, supporting the reliability of our prediction framework, we found that the trend calculated using the downscaled soil moisture predictions is closer to the trend of the ISMN data than the trend of the ESA-CCI soil moisture values (Fig. 6).

Both soil moisture gridded products (ESA-CCI and the downscaled soil moisture predictions based on digital terrain analysis

predictions) showed a variety of pixels showing both positive and negative trends during the analyzed period of time at the global scale (Fig. 7). At the global scale, from the original ESA-CCI soil moisture product we found that 25.16% of pixels (53374 of 212141 total pixels of 27x27km) with available data during 1991 and 2016 showed significant positive and negative trends (using a probability threshold of 0.05). For the soil moisture predictions based on digital terrain analysis this value showed a similar percentage (26.16%, 368614 of 1409020 total pixels) of pixels showing positive and negative trends

(same probability threshold of 0.05) during the analyzed period. The downscaled soil moisture product based on digital terrain analysis revealed a larger area of significant soil moisture decline (significant negative trend, probability threshold <0.05). The ESA-CCI soil moisture product showed 16635 pixels of 27 km with significant positive trend (12126915 km$^2$) and 36739 pixels (26782731 km$^2$) showing significant negative trend (Fig. 7a). However, in the downscaled soil moisture product a total of 103863 pixels of 15 km of spatial resolution showed a significant positive trend (23369175 km$^2$), while

264751 pixels of the same dimensions (59568975 km$^2$) showed a significant negative trend (Fig. 7b). Thus, the soil moisture decline detected using the downscaled product occupies an area >2 times larger than the area where soil moisture decline was detected using the ESA-CCI soil moisture product.

The downscaled soil moisture predictions based on digital terrain analysis are useful for quantifying soil moisture trends

across areas where no soil moisture information for long periods of time is otherwise available (Fig. 8). These soil moisture predictions revealed a negative trend across tropical rain forests of the Amazon basin in Latin America, and the Congo region in Africa (Fig. 7b). Across these areas, there are still large spatial gaps of information in the original ESA-CCI





satellite product due to intrinsic sensor limitations and therefore trends and spatial patterns cannot be resolved across these areas using the ESA-CCI satellite product on its version 4.2 (Fig. 7a) or 4.4 (Supplementary Figure S1).


## 4 Discussion

We developed a regression framework coupling ML and digital terrain analysis for improving (by nearly 50%) the spatial resolution of satellite soil moisture datasets from the ESA-CCI soil moisture product at the global scale.  We provided a gap-295   free annual mean soil moisture dataset at the global scale for years 1991-2016 using a spatial resolution of 15 km grids. This dataset could prove useful for analyzing the spatial variability of long-term drought scenarios associated with soil moisture decline (Berg and Sheffield, 2018) and its implications in global hydrological models (Zhuo et al., 2016), climate change predictions (Samaniego et al., 2018), carbon cycling models (Green et al., 2019), or for monitoring famine spatial relationships with water scarcity for crops and human use (Mishra et al., 2019).


We provided soil moisture information across areas where no information of the ESA-CCI soil moisture product is available. These results could contribute with the validation and calibration of current initiatives for improving the spatial representativeness and data quality of the ESA-CCI soil moisture product (Gruber et al., 2017). To predict soil moisture across these areas with gaps in the ESA-CCI soil moisture product (Fig. 7a) we assumed that soil moisture (each annual 305   mean soil moisture estimate) can be predicted as a function of topography (Guevara and Vargas 2019). This is based on physical principles because topography is a major control of the water distribution in the landscape and topography determines the angle between satellite soil moisture sensors and the earth surface. Thus, our results bring attention for the potential of using digital terrain parameters (i.e., surrogates of topography) for improving the spatial resolution and gap-filling of the current satellite-derived soil moisture products.


The accuracy of our downscaled soil moisture predictions (Table 3) is consistent with the accuracy of the original ESA-CCI soil moisture product (Table 2). However, we found that the extreme values (minimum and maximum values) in the ISMN dataset in an annual basis are underestimated by the ESA-CCI soil moisture product (Fig. 4). This subestimation, as explained in previous work (McColl et al., 2017, Liu et al., 2019), is because satellite soil moisture sensors are not able to 315   provide accurate estimates across extremely dry conditions or across areas where water aboveground is higher than water belowground (e.g., extremely humid conditions). For a better calibration and understanding the main limitations of satellite soil moisture measurements across multiple environmental conditions, there is an increasing number of studies reporting validation performances across multiple scales (spatial and temporal) of available soil moisture satellite datasets (An et al., 2016; Colliander et al., 2017b; Dorigo et al., 2011b; Minet et al., 2012; Mohanty et al., 2017; Yee et al., 2016). Several 320   reports have shown similar prediction bias (e.g., similar RMSE values) (Al-Yaari et al., 2019; Colliander et al., 2017a) as the results of this study. Thus, we contribute with a modeling framework supported by the direct correlation between topography



and satellite soil moisture (Mason et al., 2016) that brings positive implications to increase the accuracy of satellite soil moisture measurements.

Increasing the accuracy of satellite soil moisture measurements is critical for improving interpretations about soil moisture spatial and temporal dynamics. Thus, multiple alternative modeling evaluation frameworks and model evaluation statistics are required (or could be useful) to better interpret the spatial variability and dynamics of global soil moisture (McColl et al., 2017). The use of multiple evaluation statistics is important because there is not a single best measure, and it is necessary to use a combination of these performance measures for model evaluation purposes (Chang and Hanna, 2004). This is

specifically important when working with limited spatial data for validation of global models based on coarse scale soil moisture gridded measurements.

Our model evaluation is based on a set of multiple evaluation statistics that allow a better understanding of the discrepancies and the sources of the discrepancies between the ISMN and both soil moisture gridded datasets (ISMN vs ESA-CCI and ISMN

vs the downscaled soil moisture predictions). The sources of discrepancies can be associated with the spatial representation of multiple soil moisture products (e.g., points vs grids) and their spatial representativeness (Nicolai-Shaw et al., 2015). While the ESA-CCI product shows a coarser spatial resolution and large areas with no available data (Fig. 2), then the ISMN shows a sparse distribution of available data. In addition, most of the ISMN datasets are located at higher latitudes, but large areas around the tropics and water-limited environments are not represented within the ISMN (Fig. 1). Water limited environments

across arid and semi-arid regions where drying trends are prevalent, large discrepancy has been found between satellite and model-based soil moisture estimates (Liu et al., 2019). The lack of field soil moisture information across these areas is a major limitation for interpreting the discrepancies between satellite and model-based soil moisture estimates. This lack of information is also a limitation for interpreting the accuracy of our prediction framework (and the accuracy of the ESA-CCI soil moisture product) across large areas where no field data for validation purposes are available in the ISMN.


Using the available data contained in the ISMN for validating the ESA-CCI soil moisture product and the soil moisture predictions generated here, we found consistent negative trends of soil moisture at the global scale (Fig. 6). The main attribution of this soil moisture drying at the global scale, which is consistent with recent soil moisture monitoring efforts (Gu et al., 2019a) is anthropogenic climate change (i.e., land use change, Gu et al., 2019b). This is consistent with previous reports that have

found similar results reporting a dominance in previous decades of negative soil moisture trends across the world that were detected using field and satellite soil moisture measurements (Albergel et al., 2013). It has been shown how soil moisture decline can be intensified by land warming (Samaniego et al., 2018) or by land use change (Chen, et al., 2016, Garg et al., 2019) and agricultural practices (Bradford et al., 2017) or transformations to vegetation cover that directly affect primary productivity, evapotranspiration rates and drought (Stocker et al, 2019, Martens et al., 2018). Areas with high rates of primary

productivity and the evapotranspiration rates such as the tropical rain forest of the Amazon or the Congo regions, are examples





of areas where is challenging to accurately assess soil moisture trends due to limitations of historical soil moisture records (such as in the ESA-CCI). The ESA-CCI also lack spatial information in higher latitudes across areas with high density of small water bodies (i.e., northeast United States) or high latitude forested areas (i.e., boreal forest) constantly covered by snow (Reich et al., 2018). It has been shown that soil moisture regulates climate warming effects on forest tree species across the

aforementioned areas (Reich et al., 2018). The response of vegetation productivity to long term soil moisture trends is needed for improved land management and land surface modeling across all climate conditions and finer spatial grids. Therefore, we provide annual soil moisture values across the world (Fig. 8) that can be used to monitor the long term response of vegetation to soil moisture decline based on geomorphometry and remote sensing of soil moisture.

Our prediction framework was useful to improve the spatial representation of ESA-CCI soil moisture product. Recent soil moisture remote sensing products (Entekhabi et al., 2010, Piles et al., 2019) are able to provide soil moisture information across areas with spatial gaps in the ESA-CCI and provide global estimates, however there are only recent records with full soil moisture coverage (e.g., 2010 to date, ) and this represent a limitation for studying historical soil moisture and atmosphere interactions across longer periods of time. Soil moisture trends are crucial in future projections of the water cycle for identifying

regions of strong land–atmosphere coupling (Lorenz et al., 2015). However higher resolution of soil moisture products are needed to precisely quantify the contribution of soil moisture on larnd-surface models (Singh et al., 2015). Although model-based estimates of soil moisture associated to global land data assimilation systems are available for studying historical soil moisture trends at the global scale (Fang et al., 2009, Liu et al., 2019), they remain represented with spatial resolutions > 15km grids. By increasing the spatial resolution of the ESA-CCI soil moisture product by nearly 50%, we demonstrate the potential

of digital terrain analysis to predict satellite soil moisture spatial patterns and trends while increasing the agreement between satellite and field soil moisture records predicting historical soil moisture trends (Fig 6). While our results are consistent with previous studies predicting global soil moisture decline (Jung et al., 2010, Albergel et al., 2013), they are also generalizable to specific spatial extents or higher spatial resolution (e.g., across the continental United States using 1x1km grids, Guevara and Vargas, 2019) to analyze spatial and temporal trends of soil moisture.


Future improvements for our approach could include predicting soil moisture patterns across finer pixel sizes (e.g., 1km or <1km) and higher temporal resolutions (e.g., monthly, daily). The current version of the downscaled soil moisture predictions based on digital terrain analysis is provided in an annual basis because is a temporal resolution useful for multiple ecological and hydrological studies related to climate change (Green et al., 2019). We recognize that there is an

increasing need of soil moisture datasets with higher temporal resolutions to analyze the seasonal and short-term memory soil moisture effects after precipitation events (McColl et al., 2017). A spatial resolution of 15 km is still a coarse pixel size for detailed analysis of hydro-ecological patterns (e.g., at the hillslope scale), but the main focus of this study was to test the potential of digital terrain analysis for increasing the spatial resolution of the original ESA-CCI soil moisture product. Our



decision for selecting a 15km pixel size was driven by the reproducibility or our framework by multiple groups without the
need of high performance computing infrastructure.

In conclusion, to downscale (i.e., increase spatial resolution) coarse satellite soil moisture grids we used ML to combine
satellite soil moisture data with terrain parameters (as surrogates of topographic variability). Our results support that digital
terrain analysis can be used for improving the spatial resolution (from 27 to 15 km grids) of available global satellite soil
moisture datasets based on multiple evaluation metrics against field soil moisture data. We provide a new gap-free and
annual soil moisture product (1991-2016) that can be used for further ecological and hydrological analysis. The current
version of the new generated data set is composed by 26 annual soil moisture predictions provided across 15 km grids in an
annual basis (1991-2016). These grids could be useful for identifying and characterizing long term patterns in the soil
moisture content across 26 years of available satellite soil moisture datasets from the regional to the global scale. Future
efforts could apply our framework to address the increasing need of soil moisture datasets with higher temporal and spatial
resolution at the global scale using similar or different satellite-derived soil moisture products.

## 5 Data Source and Scientific Replicability

The training soil moisture dataset used in this study is available (here: https://www.esa-soilmoisture-cci.org/) thanks to the
ESA-CCI soil moisture initiative (Dorigo et al., 2017). The downscaled soil moisture predictions based on digital terrain
analysis are provided (Guevara, et al., 2019, https://doi.org/10.4211/hs.b940b704429244a99f902ff7cb30a31f) in a set raster
files with a *.tif extension for generic raster formats (e.g., 1 raster per year, folder: predicted_sm_global_15km) . This rasters
(n=26, 1991-2016) can be imported to any GIS.

To ensure the replicability of this study we also provide a spatial data frame (in R native format *.rds) with the topographic
terrain parameters (e.g., file: topographic_predictors_15km_grids.rds, also provided as *.tif raster in folder:
prediction_factors_15km) used as prediction factors for the yearly means of the ESA-CCI soil moisture product. We provide
the yearly means of the ISMN database that we used for evaluating the aforementioned soil moisture predictions. The ISMN
yearly means are harmonized with the ESA-CCI soil moisture product and the downscaled soil moisture predictions based
on terrain analysis, and provided in two separated files (e.g., files: harmonizedISMNvsESACCI.rds and
harmonizedISMNvsPREDICTED.rds). These predictions are based on ML and digital terrain parameters and they are
provided in a generic raster format (Guevara, et al., 2019, https://doi.org/10.4211/hs.b940b704429244a99f902ff7cb30a31f).
Finally, we provide the processing R code used to develop our prediction framework (e.g., file:
prediction_kknn_sm_terrain_global_15km_v0.R).

**Acknowledgements**



MG acknowledges support from a CONACYT doctoral fellowship (382790). RV and MT acknowledge support from the National Science Foundation grant CIF21 DIBBs: PD: Cyberinfrastructure Tools for Precision Agriculture in the 21st

Century. We thank to Anita Z. Schwartz from the University of Delaware for her assistance preparing the global terrain dataset. We thank Ricardo Llamas from the University of Delaware for preparing Supplementary Figure S1.

**Author contributions**

MG, RV and MT conceptualized the project. MG performed analysis and wrote the first manuscript draft. This draft was
revised, commented and edited by RV, MG and MT.

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



**Table Legends:**

- Table 1: Cross validated correlation (r), RMSE, training data pixels (n), the kernel type, and the number of neighbors of the parameter k in the soil moisture prediction models for each year.

- Table 2: Agreement metrics between the ISMN dataset and the ESA-CCI soil moisture product at the annual scale. n, the number of complete pairs of data; FAC2, fraction of predictions within a factor of two; MB, the mean bias; MGE, the mean gross error; NMB, the normalized mean bias; NMGE, the normalized mean gross error; RMSE, the root mean squared error; r, the Pearson correlation coefficient; COE, the Coefficient of Efficiency; IOA, the Index of Agreement.

- Table 3: Agreement metrics between the ISMN dataset and the downscaled soil moisture predictions based on digital terrain analysis. n, the number of complete pairs of data; FAC2, fraction of predictions within a factor of two; MB, the mean bias; MGE, the mean gross error; NMB, the normalized mean bias; NMGE, the normalized mean gross error; RMSE, the root mean squared error; r, the Pearson correlation coefficient; COE, the Coefficient of Efficiency; IOA, the Index of Agreement.




**Table 1: Cross validated correlation (r), RMSE, training data pixels (n), the kernel type, and the number of neighbors of the parameter k in the soil moisture prediction models for each year.**


|    | year | r    | rmse   | n      | kernel     | k  |
|----|------|------|--------|--------|------------|----|
| 1  | 1991 | 0.78 | 0.0448 | 195886 | triangular | 20 |
| 2  | 1992 | 0.81 | 0.0404 | 198624 | triangular | 19 |
| 3  | 1993 | 0.81 | 0.0405 | 198144 | triangular | 19 |
| 4  | 1994 | 0.81 | 0.0408 | 198458 | triangular | 19 |
| 5  | 1995 | 0.82 | 0.0400 | 198595 | triangular | 18 |
| 6  | 1996 | 0.81 | 0.0398 | 198752 | triangular | 18 |
| 7  | 1997 | 0.81 | 0.0398 | 198655 | triangular | 19 |
| 8  | 1998 | 0.81 | 0.0409 | 199076 | triangular | 18 |
| 9  | 1999 | 0.81 | 0.0407 | 199067 | triangular | 18 |
| 10 | 2000 | 0.81 | 0.0404 | 199089 | triangular | 19 |
| 11 | 2001 | 0.80 | 0.0422 | 199008 | triangular | 19 |
| 12 | 2002 | 0.81 | 0.0407 | 208167 | triangular | 19 |
| 13 | 2003 | 0.78 | 0.0441 | 178580 | triangular | 21 |
| 14 | 2004 | 0.83 | 0.0374 | 152696 | triangular | 19 |
| 15 | 2005 | 0.82 | 0.0398 | 173179 | triangular | 18 |
| 16 | 2006 | 0.80 | 0.0405 | 176761 | triangular | 19 |
| 17 | 2007 | 0.80 | 0.0398 | 208161 | triangular | 18 |
| 18 | 2008 | 0.81 | 0.0393 | 209543 | triangular | 18 |
| 19 | 2009 | 0.81 | 0.0391 | 209276 | triangular | 19 |
| 20 | 2010 | 0.79 | 0.0398 | 211717 | triangular | 18 |
| 21 | 2011 | 0.80 | 0.0391 | 211581 | triangular | 19 |
| 22 | 2012 | 0.80 | 0.0399 | 212042 | triangular | 18 |
| 23 | 2013 | 0.81 | 0.0400 | 209584 | triangular | 18 |
| 24 | 2014 | 0.80 | 0.0403 | 209573 | triangular | 18 |
| 25 | 2015 | 0.80 | 0.0402 | 209590 | triangular | 19 |
| 26 | 2016 | 0.80 | 0.0402 | 209711 | triangular | 19 |





**Table 2: Agreement metrics between the ISMN dataset and the ESA-CCI soil moisture product at the annual scale.**
**n, the number of complete pairs of data;  FAC2, fraction of predictions within a factor of two;  MB, the mean bias;**
**MGE, the mean gross error;  NMB, the normalized mean bias; NMGE, the normalized mean gross error; RMSE, the**
**root mean squared error; r, the Pearson correlation coefficient; COE, the Coefficient of Efficiency; IOA, the Index of**
**Agreement.**

|    | year | n    | FAC2  | MB     | MGE   | NMB    | NMGE  | RMSE  | r     | COE    | IOA   |
|----|------|------|-------|--------|-------|--------|-------|-------|-------|--------|-------|
| 1  | 1991 | 102  | 0.892 | -0.069 | 0.081 | -0.241 | 0.282 | 0.105 | 0.091 | -0.499 | 0.251 |
| 2  | 1992 | 123  | 0.837 | -0.077 | 0.091 | -0.269 | 0.316 | 0.115 | 0.165 | -0.455 | 0.272 |
| 3  | 1993 | 113  | 0.903 | -0.076 | 0.085 | -0.251 | 0.282 | 0.108 | 0.234 | -0.466 | 0.267 |
| 4  | 1994 | 113  | 0.823 | -0.088 | 0.093 | -0.292 | 0.308 | 0.119 | 0.055 | -0.782 | 0.109 |
| 5  | 1995 | 94   | 0.851 | -0.078 | 0.090 | -0.272 | 0.317 | 0.110 | 0.251 | -0.478 | 0.261 |
| 6  | 1996 | 118  | 0.847 | -0.099 | 0.106 | -0.332 | 0.357 | 0.128 | 0.252 | -0.644 | 0.178 |
| 7  | 1997 | 121  | 0.909 | -0.080 | 0.098 | -0.274 | 0.339 | 0.115 | 0.319 | -0.433 | 0.284 |
| 8  | 1998 | 119  | 0.899 | -0.082 | 0.098 | -0.282 | 0.334 | 0.117 | 0.357 | -0.380 | 0.310 |
| 9  | 1999 | 72   | 0.903 | -0.055 | 0.091 | -0.203 | 0.337 | 0.103 | 0.589 | -0.012 | 0.494 |
| 10 | 2000 | 82   | 0.927 | -0.062 | 0.089 | -0.228 | 0.328 | 0.104 | 0.522 | -0.067 | 0.467 |
| 11 | 2001 | 133  | 0.880 | -0.039 | 0.085 | -0.163 | 0.357 | 0.102 | 0.465 | 0.077  | 0.538 |
| 12 | 2002 | 205  | 0.883 | -0.038 | 0.078 | -0.168 | 0.349 | 0.094 | 0.581 | 0.126  | 0.563 |
| 13 | 2003 | 299  | 0.796 | -0.032 | 0.083 | -0.148 | 0.385 | 0.102 | 0.470 | 0.108  | 0.554 |
| 14 | 2004 | 380  | 0.845 | -0.047 | 0.081 | -0.205 | 0.351 | 0.105 | 0.433 | 0.041  | 0.521 |
| 15 | 2005 | 469  | 0.806 | -0.033 | 0.083 | -0.156 | 0.390 | 0.109 | 0.271 | 0.020  | 0.510 |
| 16 | 2006 | 507  | 0.805 | -0.033 | 0.080 | -0.159 | 0.381 | 0.102 | 0.308 | 0.008  | 0.504 |
| 17 | 2007 | 573  | 0.859 | -0.028 | 0.073 | -0.133 | 0.350 | 0.090 | 0.452 | 0.059  | 0.529 |
| 18 | 2008 | 613  | 0.853 | -0.041 | 0.081 | -0.187 | 0.368 | 0.106 | 0.453 | 0.065  | 0.532 |
| 19 | 2009 | 696  | 0.876 | -0.035 | 0.079 | -0.157 | 0.358 | 0.104 | 0.468 | 0.089  | 0.544 |
| 20 | 2010 | 900  | 0.839 | -0.029 | 0.078 | -0.135 | 0.363 | 0.107 | 0.331 | 0.041  | 0.521 |
| 21 | 2011 | 1031 | 0.826 | -0.032 | 0.080 | -0.152 | 0.375 | 0.112 | 0.334 | 0.046  | 0.523 |
| 22 | 2012 | 1318 | 0.705 | -0.045 | 0.091 | -0.218 | 0.439 | 0.126 | 0.200 | -0.055 | 0.472 |
| 23 | 2013 | 1242 | 0.812 | -0.019 | 0.076 | -0.099 | 0.387 | 0.107 | 0.433 | 0.133  | 0.566 |
| 24 | 2014 | 1237 | 0.840 | -0.026 | 0.075 | -0.126 | 0.368 | 0.103 | 0.469 | 0.129  | 0.565 |
| 25 | 2015 | 1253 | 0.812 | -0.034 | 0.083 | -0.161 | 0.388 | 0.119 | 0.415 | 0.126  | 0.563 |





| 26 | 2016 | 1165 | 0.842 | -0.035 | 0.080 | -0.163 | 0.368 | 0.117 | 0.413 | 0.108 | 0.554 |

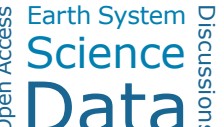

**Table 3: Agreement metrics between the ISMN dataset and the downscaled soil moisture predictions based on digital terrain analysis. n, the number of complete pairs of data; FAC2, fraction of predictions within a factor of two; MB, the mean bias; MGE, the mean gross error; NMB, the normalized mean bias; NMGE, the normalized mean gross error; RMSE, the root mean squared error; r, the Pearson correlation coefficient; COE, the Coefficient of Efficiency; IOA, the Index of Agreement.**


|   | year | n | FAC2 | MB | MGE | NMB | NMGE | RMSE | r | COE | IOA |
|---|------|---|------|----|----|-----|------|------|---|-----|-----|
| 1 | 1991 | 104 | 0.904 | -0.078 | 0.084 | -0.270 | 0.292 | 0.108 | 0.077 | -0.561 | 0.220 |
| 2 | 1992 | 125 | 0.880 | -0.084 | 0.094 | -0.292 | 0.326 | 0.118 | 0.120 | -0.509 | 0.245 |
| 3 | 1993 | 115 | 0.887 | -0.085 | 0.093 | -0.280 | 0.306 | 0.116 | 0.112 | -0.597 | 0.202 |
| 4 | 1994 | 115 | 0.826 | -0.092 | 0.095 | -0.302 | 0.315 | 0.117 | 0.133 | -0.806 | 0.097 |
| 5 | 1995 | 96 | 0.865 | -0.085 | 0.094 | -0.296 | 0.328 | 0.115 | 0.214 | -0.528 | 0.236 |
| 6 | 1996 | 119 | 0.849 | -0.099 | 0.106 | -0.333 | 0.355 | 0.127 | 0.276 | -0.638 | 0.181 |
| 7 | 1997 | 122 | 0.910 | -0.081 | 0.098 | -0.279 | 0.337 | 0.117 | 0.295 | -0.427 | 0.287 |
| 8 | 1998 | 120 | 0.900 | -0.082 | 0.096 | -0.279 | 0.326 | 0.116 | 0.372 | -0.349 | 0.325 |
| 9 | 1999 | 73 | 0.904 | -0.051 | 0.092 | -0.189 | 0.344 | 0.103 | 0.601 | -0.015 | 0.492 |
| 10 | 2000 | 84 | 0.940 | -0.056 | 0.089 | -0.208 | 0.328 | 0.102 | 0.560 | -0.047 | 0.477 |
| 11 | 2001 | 136 | 0.904 | -0.031 | 0.083 | -0.129 | 0.348 | 0.096 | 0.544 | 0.106 | 0.553 |
| 12 | 2002 | 211 | 0.872 | -0.030 | 0.079 | -0.134 | 0.355 | 0.094 | 0.568 | 0.127 | 0.564 |
| 13 | 2003 | 307 | 0.798 | -0.027 | 0.084 | -0.128 | 0.391 | 0.101 | 0.492 | 0.108 | 0.554 |
| 14 | 2004 | 389 | 0.871 | -0.040 | 0.081 | -0.173 | 0.354 | 0.103 | 0.411 | 0.045 | 0.522 |
| 15 | 2005 | 483 | 0.810 | -0.032 | 0.083 | -0.150 | 0.388 | 0.107 | 0.331 | 0.031 | 0.516 |
| 16 | 2006 | 522 | 0.830 | -0.031 | 0.080 | -0.148 | 0.376 | 0.101 | 0.353 | 0.035 | 0.517 |
| 17 | 2007 | 586 | 0.884 | -0.026 | 0.073 | -0.127 | 0.351 | 0.090 | 0.454 | 0.059 | 0.530 |
| 18 | 2008 | 626 | 0.874 | -0.041 | 0.082 | -0.187 | 0.371 | 0.107 | 0.456 | 0.056 | 0.528 |
| 19 | 2009 | 710 | 0.885 | -0.034 | 0.081 | -0.154 | 0.365 | 0.106 | 0.449 | 0.070 | 0.535 |
| 20 | 2010 | 911 | 0.856 | -0.028 | 0.080 | -0.129 | 0.368 | 0.106 | 0.333 | 0.030 | 0.515 |
| 21 | 2011 | 1051 | 0.829 | -0.031 | 0.082 | -0.143 | 0.384 | 0.112 | 0.301 | 0.024 | 0.512 |
| 22 | 2012 | 1341 | 0.786 | -0.035 | 0.084 | -0.166 | 0.405 | 0.115 | 0.299 | 0.028 | 0.514 |
| 23 | 2013 | 1273 | 0.807 | -0.012 | 0.079 | -0.061 | 0.402 | 0.108 | 0.380 | 0.096 | 0.548 |
| 24 | 2014 | 1268 | 0.834 | -0.020 | 0.079 | -0.098 | 0.382 | 0.106 | 0.380 | 0.091 | 0.546 |
| 25 | 2015 | 1283 | 0.812 | -0.028 | 0.086 | -0.129 | 0.401 | 0.121 | 0.341 | 0.093 | 0.547 |





| 26 | 2016 | 1194 | 0.845 | -0.031 | 0.082 | -0.145 | 0.378 | 0.118 | 0.366 | 0.086 | 0.543 |



**Figure captions:**

- Figure 1: Data distribution of the ISMN dataset (n=13376) available for the period 1991-2016. The gray line shows geopolitical borders.

- Figure 2: ESA-CCI soil moisture mean (a) and standard deviation (b) for the period 1991-2016. White areas are areas where no complete information is available during the analyzed period. The black line shows geopolitical borders.

- Figure 3: Topographic terrain parameters that were derived from the DEM using SAGA-GIS. These terrain parameters were used as prediction factors for the values of the ESA-CCI soil moisture product. These terrain parameters were standardized by centering their means in 0 by a variance unit for improving visualization purposes. a) aspect: terrain aspect, b) carea: specific catchment area, c) chnl base: channel network base level, d) chnl dist: distance to channel network, e) convergence: flow convergence index, f) hcurv: horizontal curvature, g) land:

  digital elevation model, h) lsfactor: length-slope factor, i) rsp: relative slope position, j) shade: analytical hillshading, k) sinks: smoothed elevation, l) slope: terrain slope, m) vall depth: valley depth index, n) vcurv: vertical curvature, o) wetness: topographic wetness index.

- Figure 4: Probability distribution functions showing the statistical distribution of soil moisture gridded datasets and soil moisture observations from the ISMN.

- Figure 5: Predicted soil moisture mean based on digital terrain parameters (a) and standard deviation (b) for the period 1991-2016. The black line shows geopolitical borders.

- Figure 6: Trend detection results for soil moisture. Trends of the ISMN dataset (a), the soil moisture predictions based on digital terrain analysis (b) and the ESA-CCI soil moisture product (c).

- Figure 7: Pixel based soil moisture annual trend based on the ESA-CCI soil moisture product (a) and soil moisture
  annual trends from the soil moisture predictions based on digital terrain analysis (b) for the period 1991-2016. The black line shows geopolitical borders. Dark gray areas are areas where no significant trend (p-value>0.05) was detected.

- Figure 8. Soil moisture predicted across areas with gaps in the ESA-CCI soil moisture product. Black areas in the global map indicate the areas that are well covered by the ESA-CCI soil moisture product while the colored areas
  are the predictions of soil moisture based on geomorphometry and machine learning (a). We show soil moisture predictions across areas with no data in the ESA-CCI soil moisture product across the Amazon (b) and the Congo (c) tropical forests.




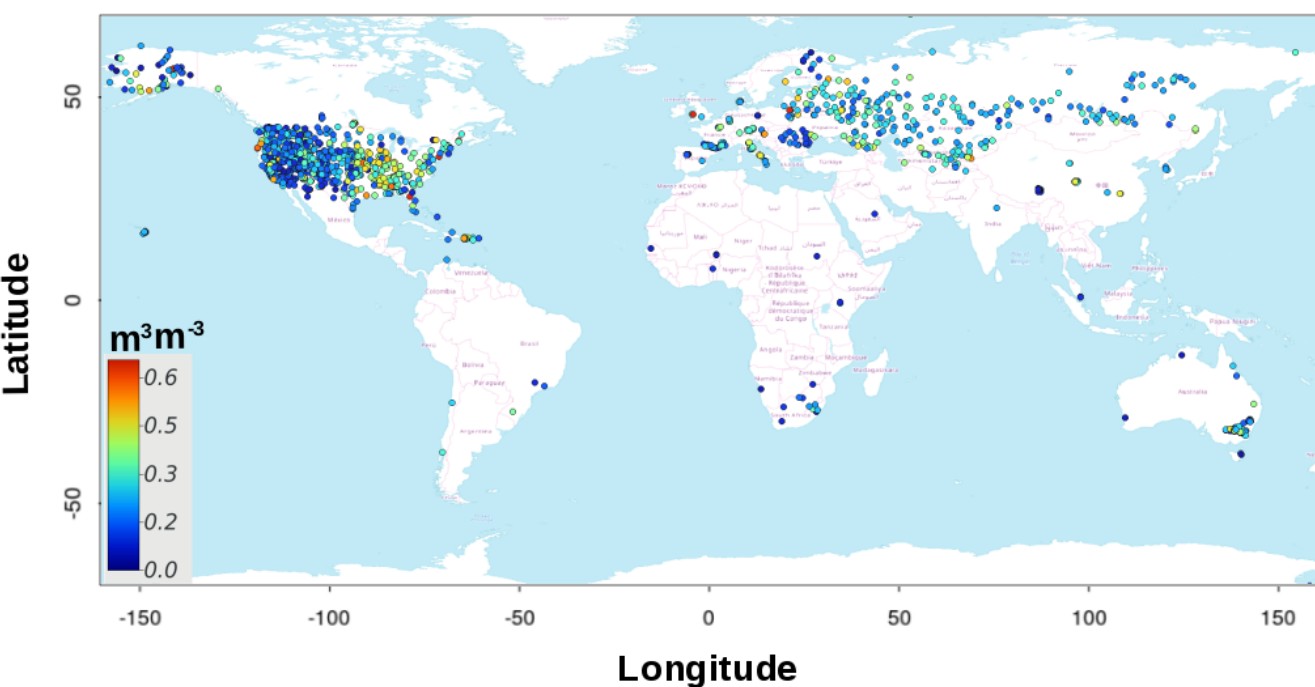

**Figure 1: Data distribution of the ISMN dataset (n=13376) available for the period 1991-2016. Values represent overall mean values at each location for the period 1991-2016.**

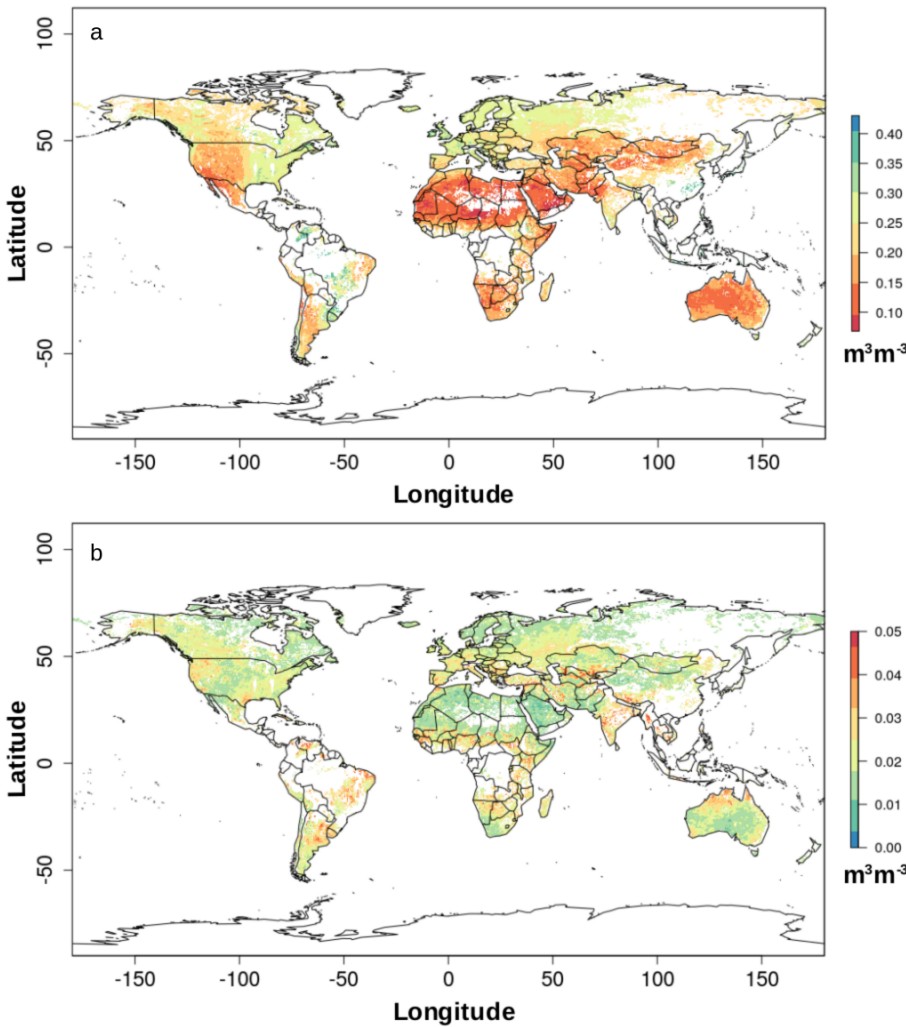

**Figure 2: ESA-CCI soil moisture mean (a) and standard deviation (b) for the period 1991-2016 from ESA-CCI soil moisture version 4.2. White areas are areas where no complete information is available during the analyzed period. The black line shows geopolitical borders.**








**Figure 3: Topographic terrain parameters that were derived from the DEM using SAGA-GIS. These terrain parameters were used as prediction factors for the values of the ESA-CCI soil moisture product. These terrain parameters were standardized by centering their means in 0 by a variance unit for improving visualization purposes. a) aspect: terrain aspect, b) carea: specific catchment area, c) chnl base: channel network base level, d) chnl dist: distance to channel network, e) convergence: flow convergence index, f) hcurv: horizontal curvature, g) land: digital**

**elevation model, h) lsfactor: length-slope factor, i) rsp: relative slope position, j) shade: analytical hillshading, k) sinks: smoothed elevation, l) slope: terrain slope, m) vall depth: valley depth index, n) vcurv: vertical curvature, o) wetness: topographic wetness index.**







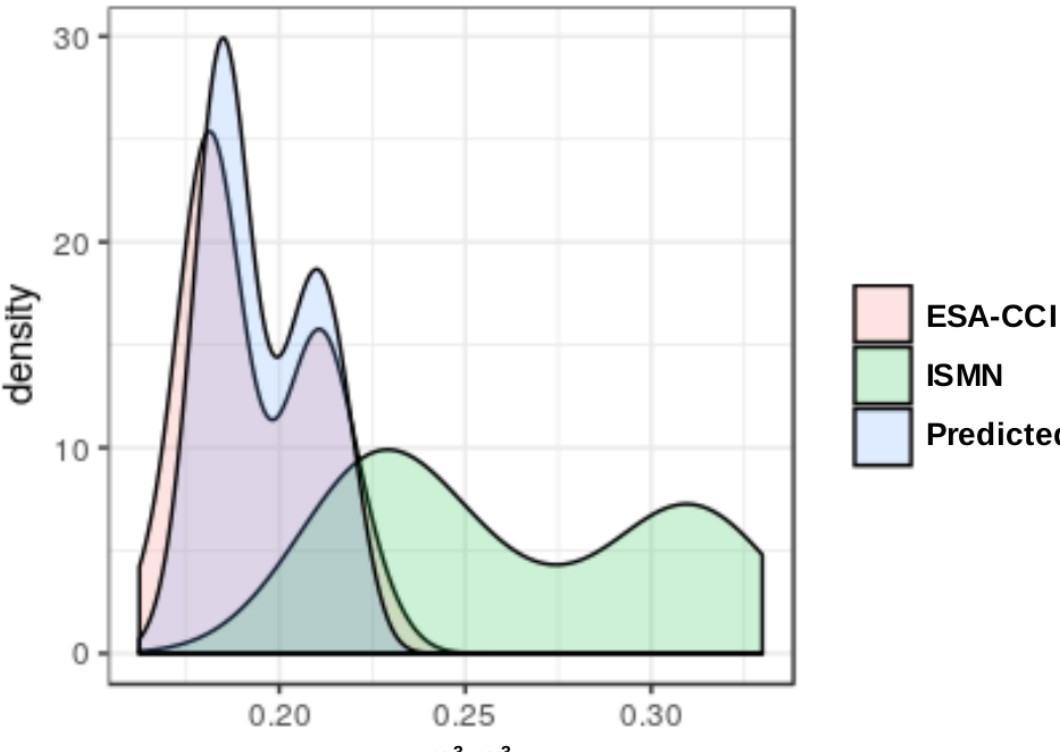


**Figure 4: Probability distribution functions showing the statistical distribution of soil moisture gridded datasets (i.e., ESA-CCI, and our soil moisture product [Predicted]) and soil moisture observations from the ISMN.**


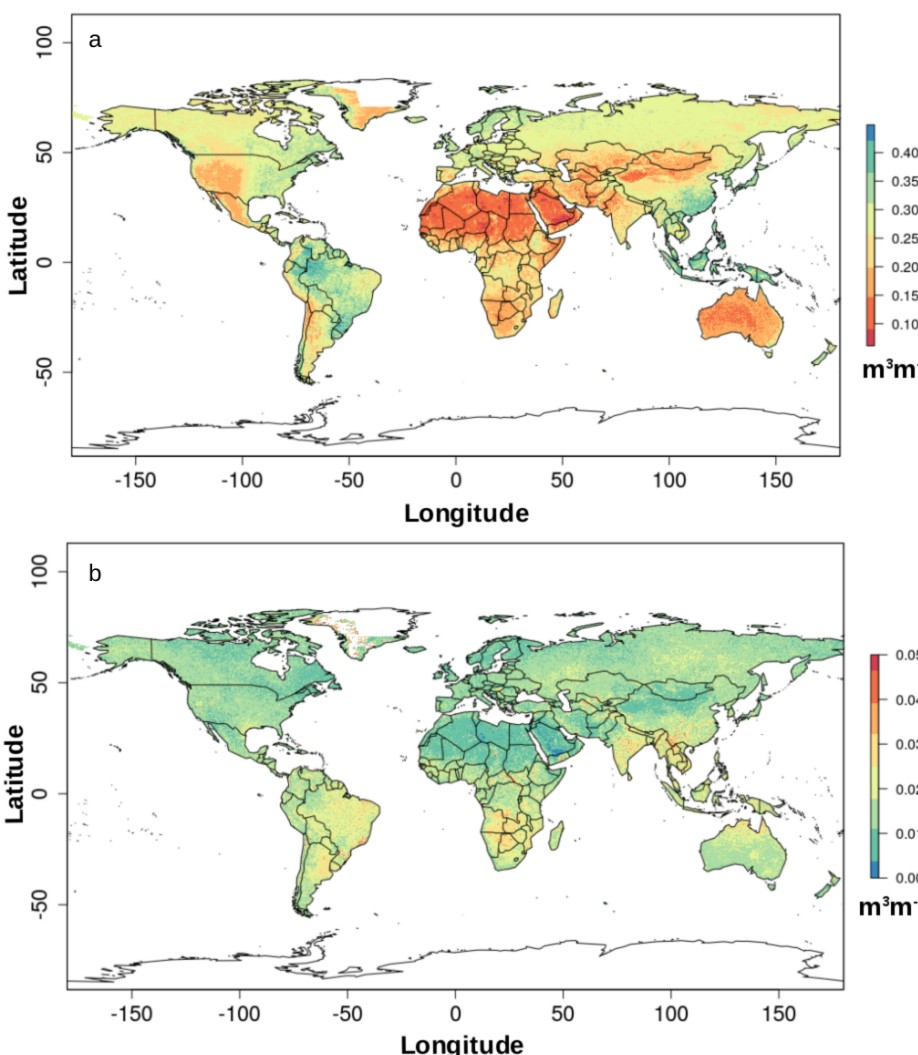

**Figure 5: Predicted soil moisture mean based on digital terrain parameters (a) and standard deviation (b) for the period 1991-2016. The black line shows geopolitical borders.**

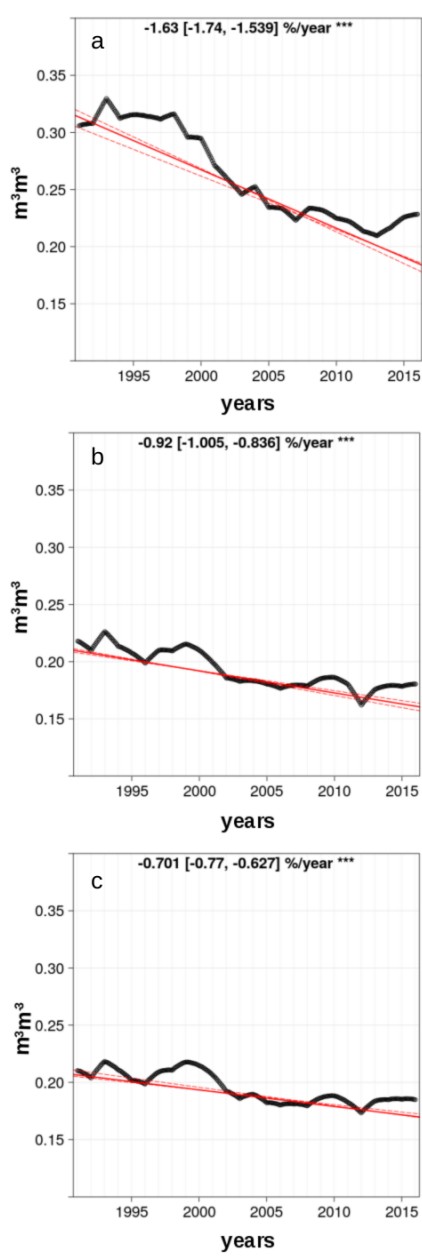

**Figure 6: Trend detection results for soil moisture. Trends of the ISMN dataset (a), the soil moisture predictions based on digital terrain analysis (b) and the ESA-CCI soil moisture product (c).**

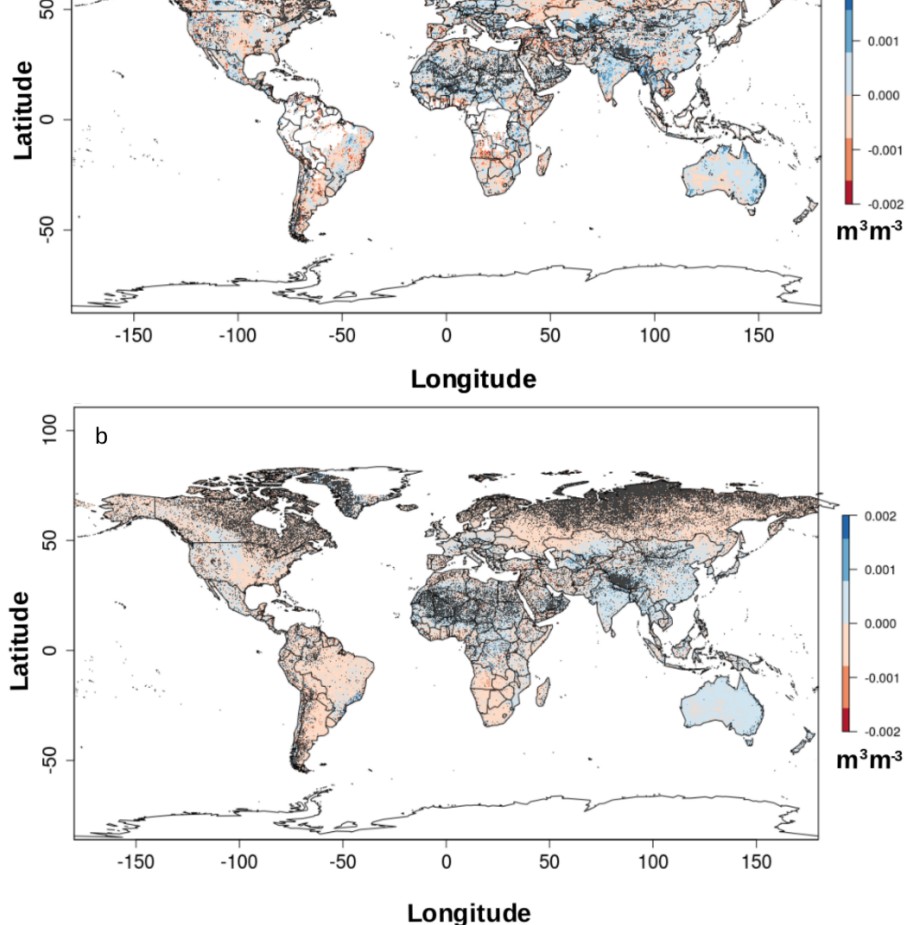

**Figure 7: Pixel based soil moisture annual trend based on the ESA-CCI soil moisture product (a) and soil moisture annual trends from the soil moisture predictions based on digital terrain analysis (b) for the period 1991-2016. The black line shows geopolitical borders. Dark gray areas are areas where no significant trend (p-value >0.05) was detected.**

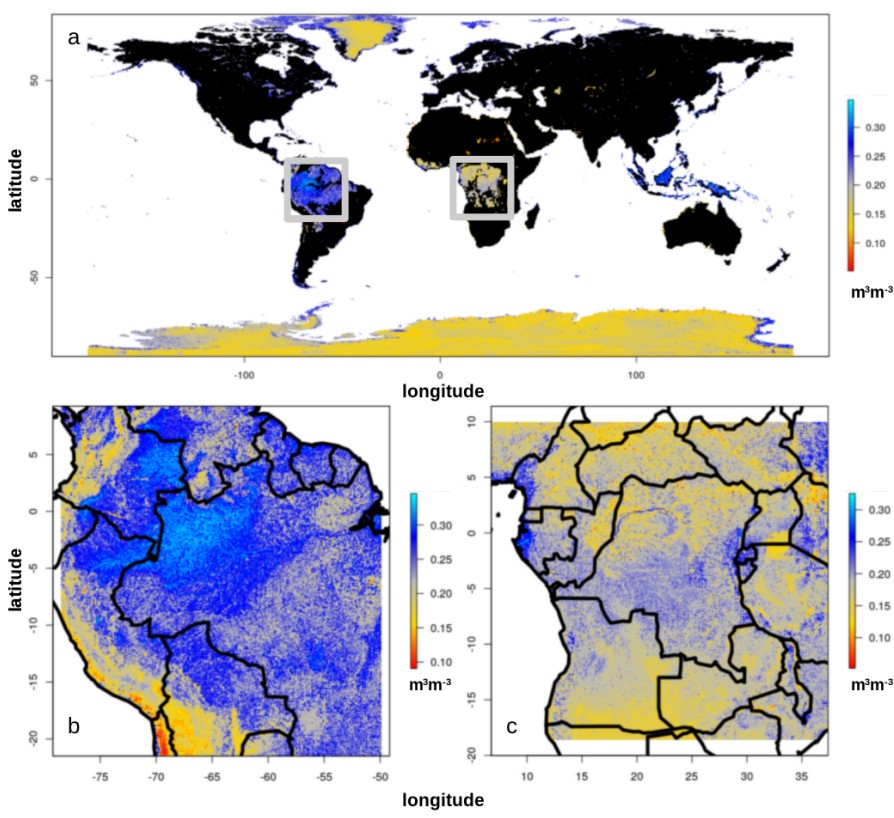

**Figure 8. Soil moisture predicted across areas with gaps in the ESA-CCI soil moisture product. Black areas in the global map indicate the areas that are well covered by the ESA-CCI soil moisture product while the colored areas are the predictions of soil moisture based on geomorphometry and machine learning (a). We show mean annual soil moisture predictions across areas with no data in the ESA-CCI soil moisture product across the Amazon (b) and the Congo (c) basins.**