# Peer review of "Gap-Free Global Annual Soil Moisture: 15km Grids for 1991-2016"

_Earth System Science Data, 2019_

## Referee Comment (RC1) · Anonymous Referee #1 · 8 Nov 2019

This paper presents a 15 km annual-average soil moisture product that is generated by machine-learning the relation between 0.25 degree ESA CCI soil moisture estimates and topographic indices derived from a higher-resolution DEM.

I have several major concerns regarding the hypothesis/assumptions on which the methodology is based as well as the employed validation methodology, and consequently also the conclusions drawn from the presented analysis:

The methodology is based on the hypothesis that topography is a main driving factor for soil moisture patterns. However, the reference used to support this claim (Mason et al., 2016) presents only a very local analysis of differences between soil moisture values at low-slope and high-slope areas over grasslands only, and only in a small region over the UK. The observed relation is relatively low ($R^2 = 0.21$) and the authors

conclude "[...] a topographic signal can be seen in high resolution remotely sensed surface soil moisture data [...]. Unfortunately this signal is relatively weak." Moreover, Mason et al. (2016) uses 1 km SAR data for which topographic corrections are applied in the pre-processing, which likely induces a sensitivity of the measurements to topography parameters. These topography corrections are usually not applied to coarse-resolution measurements such as the sensors used within the ESA CCI SM, because topographic effects average out at these scales. The presented paper itself also does not analyze the predictive power of the used topographic indices for soil moisture (e.g., the godness-of-fit for the obtained regression, variable importance, etc.).

Hence, there is no evidence supporting the reliability of topographic indices as predictor for soil moisture, especially on a global scale. Even more doubtable is the assumption that the developed regression function can be used to extrapolate soil moisture to regions not covered by the ESA CCI SM, which are mainly the arctic ice sheet and tropical forests. Tropical rainforests, for example, have a quite unique moisture regime that is expected to be largely rainfall dependent. It is very questionable to use a soil moisture - topography relation that is trained over non-tropical regions to predict soil moisture there. Moreover, no in situ measurements are available in these regions to verify the validity of these predictions.

Also, the presented validation does not support the conclusions. First, the statement (L234) "In all cases, the evaluation statistics are equal or better for the downscaled soil moisture predictions based on digital terrain analysis (Table 3) than the original ESA-CCI soil moisture product (Table 2)" is wrong. In fact, results are quite balanced, sometimes the downscaled product is "better", sometimes the original is "better", but most likely results are not actually distinguishable within reasonable confidence limits (which should be estimated). The authors do indeed acknowledge (L252): "The downscaled predictions based on digital terrain analysis are not significantly different compared with the ESA-CCI soil moisture product [...]", but the subsequent conclusions are not supported. Specifically, "[...] but they provide (1) gap free soil moisture-related

information ": while they are provided, there is no evidence that they are of any reasonable accuracy (for the earlier discussed regions), and "(2) higher resolution (from 27 to 15 km grids)": This is a mix-up of resolution and sampling.

Improved resolution would imply that there is different / more information in the downscaled product, but the indistinguishability of performance metrics (see above) suggests that this is not the case. Also the comparison in Figure 6 (b and c) shows that the original and the downscaled products exhibit the exact same behaviour with a slightly lower overall variability in the original product. It is, however, not clear whether this different overall magnitude reflects an actual improvement, because soil moisture varaibility and trends are actually supposed to be different at a point scale and at a satellite scale (see e.g. Famiglietti et al. 2008). Also, the soil moisture mean is supposed to be different at different scales, hence the negative bias between point and satellite measurements cannot be reliably interpreted as error, and a reduction of this bias may as well be a going in the wrong direction with respect to the true areal-average mean.

In other words, even though the generated product is sampled on a higher-resolution grid, it can not be concluded that this product contains higher-resolution information. Given the low amount of evidence that topography (alone) is a good predictor for soil moisture, observed differences may well be a result of the smoothing-nature of the KKNN approach, and any spatial-window resampling approach may lead to a seemingly "higher-resolution" (which is truly only a higher-sampling) product with the same (or even better) performance, but this is not tested.

Therefore, I recommend to reject this publication. However, I do believe that topography may well be an important complementary predictor for soil moisture at higher-resolution when combined with other dominant factors. I therefore encourage the authors to pursue this approach addressing the concerns outlined above.

References: Famiglietti, J. S., Ryu, D., Berg, A. A., Rodell, M., & Jackson, T. J. (2008).

Field observations of soil moisture variability across scales. Water Resources Research, 44(1). https://doi.org/10.1029/2006WR005804

---

## Referee Comment (RC2) · Anonymous Referee #2 · 11 Nov 2019

**OVERVIEW**

The study has developed a gap-filled, and downscaled with topography-derived information, long-term annual (15 km) global soil moisture dataset based on the ESA CCI satellite soil moisture products. The assessment of the dataset with respect to in situ observations has been carried out through annual comparisons as well as in terms of long-term trends (1991-2016).

**GENERAL COMMENTS**

The paper is mostly well written and clear. The topic of the paper is interesting for the readership of Earth System Science Data as a global scale gap-filled annual soil moisture dataset is surely useful for many applications. However, I believe the paper needs

major changes before the publication as several parts are not properly described and other sections need to be improved or summarized. I have listed below my comments with the indication of their relevance.

1) MAJOR: The factors used in the downscaling and gap-filling algorithm should be described in details. Two figures (Figures 2 and 3) are not mentioned in the text. It seems a part is missing. The reader needs to know the details on the methodology employed and which factors have been found to be more important. Which Digital Elevation Model is used? Additionally, other static factors such as vegetation and soil types are not considered. Why? The discussion on the approach employed for downscaling and gap-filling needs to be included in the paper. Why do the authors select such approach?

2) MAJOR: I have found particularly challenging performing gap-filling over dense forest regions in the Amazon and in Congo. Satellite soil moisture data cannot be used in such regions due to dense forest that mask the soil moisture signal. How is it possible to extend the signal there only based on topography? I would suggest to perform a more detailed validation in these areas. I strongly suggest to perform a comparison with modelled datasets (e.g., ERA5 soil moisture) to have an assessment of the performance over dense vegetated areas. A similar comment can be done for high latitude areas in which frozen soils and snow completely mask the soil moisture signal. Please perform a detailed validation over these areas, too.

3) MAJOR: The trend analysis is very interesting. However, as above, we need more details on how trends are computed. For instance, in situ stations are available only at some points over the Earth, are the same locations used with the satellite-derived datasets? If not, the comparison is wrong. Similarly, in situ stations are not available every year, and for the full year. How are the data aggregated in time and space? These details are needed. I expect the results are strongly impacted to these choices.

4) MODERATE: The machine learning downscaling approach provides soil moisture

data with a resolution higher than the original ESA CCI product. However, I am always doubtful on these downscaling approaches as instead of resolution it should be higher spatial sampling. The higher spatial resolution should be tested, but I am aware it is very hard to do (I have this comment for all downscaling studies). The authors should demonstrate that the downscaled product is able to reproduce features at higher resolutions with respect to the parent ESA CCI product. It is not done in the paper, that's why I believe higher spatial sampling, and not spatial resolution, is more appropriate.

5) MODERATE: Several performance scores have been used in the paper. However, I don't think it is necessary to use all of them. The authors should discuss what information each performance score is providing for the assessment of the dataset, not simply to list many numbers. Indeed, Tables 2 and 3 are hard to read and not informative. Please summarize only the more relevant scores in a figure.

6) MAJOR: The range of values of ESA CCI soil moisture products has little value, as the satellite products are rescaled to match the range of variability of modelled soil moisture from GLDAS. Therefore, the range of values is that obtained from GLDAS. For the analysis shown in Figure 4, and similarly for the trend analysis, the soil moisture datasets should be rescaled between the minimum and maximum of each time series and expressed as relative soil moisture (between 0 and 1). Then the data should be aggregated and the range of values and the trends can be assessed.

7) MODERATE: I believe the discussion section must be rewritten. General results are mostly discussed, whereas it should be closely related to the results shown in the paper. I believe it should be shorter and better focused.

**SPECIFIC COMMENT (L: line or lines)**

L307: Why the "angle between satellite sensors and the earth surface" is useful for determining soil moisture? It has no sense and I believe it is wrong.

**RECOMMENDATION**

Based on the above comments, I suggest a major revision before the possible publication on Earth System Science Data.

---

## Referee Comment (RC3) · Anonymous Referee #3 · 25 Nov 2019

General comments: While seeking a higher resolution global soil moisture product is certainly a laudable goal, I find the methods used in this paper lack a credible connection between ground measurements and the remotely sensed data. The authors use machine learning to generate regressions between multiple topographic variables thought to be related to soil moisture that are available at higher resolution with coarse resolution satellite data. It is difficult for the reader to discern if actual new information results from the downscaling because there is not a clear connection made between in-situ soil moisture measurements, their physical connection to the chosen topographic variables used, and resultant satellite measurements. To do so, requires first demonstrating the rigor of the methodology over a much smaller and better measured area, such as the area of the International Soil Moisture Network. Extending the algorithm to

areas that are hydrogeomorphically and climatically distinct from the existing soil moisture measurement networks cannot be result in credible data. I do not recommend publication of this work.

Major comment: Much more information is needed about the regression methods and parameter selection. It would be helpful to discuss what each parameter is mathematically and why it can be useful in a soil moisture prediction context. Can it be demonstrated over smaller areas that there is a valid argument for using these variables, some or all of them. Explain in detail the cross validation process used and why. It is unclear if there is a single model developed and applied to all years or separate models for each year. Are all topographic parameters used in the model in all years? What is their weighting?

Major comment: Extending the results to areas without soil moisture measurements (whether in situ or remotely sensed values) without any validation is not an improvement over current spatial data. It is highly suspect and could lead to inappropriate applications using what is in effect non-data.

Moderate comment: I found the paper very repetitive and in need of detailed editing.

---

## Short Comment (SC1) · 31 Dec 2019

REVIEWER COMMENT: This paper presents a 15 km annual-average soil moisture product that is generated by machine-learning the relation between 0.25 degree ESA CCI soil moisture estimates and topographic indices derived from a higher-resolution DEM. I have several major concerns regarding the hypothesis/assumptions on which the methodology is based as well as the employed validation methodology, and consequently also the conclusions drawn from the presented analysis:

AUTHORS RESPONSE: We appreciate the reviewer comments as they provide valuable feedback to increase the impact of our work. We revised our work and will provide analyzes (including new datasets and analyzes of variable importance) to improve the

validation methodology. We have already started to update our work with new datasets (e.g., eco-climatic and soil type classes) and updated our results with the recently released version of the ESA-CCI soil moisture product (4.5, up to 2018). These results will be uploaded in a fully revised version.

To increase reproducibility of our prediction framework we have started compiling an R program (that will be available in a revised version) able to predict soil moisture based on remote sensing and machine learning, from the global to the country specific scales. This new code is able to perform a bootstrapping approach given a user defined sample size (if not specified, it will use 1/3 of available data/pixels for each year) to analyze the variance of model predictions as a function of variations in training data. Thus in a revised version we will report a spatial explicit metric of model-based uncertainty.

REVIEWER COMMENT: The methodology is based on the hypothesis that topography is a main driving factor for soil moisture patterns. However, the reference used to support this claim (Mason et al., 2016) presents only a very local analysis of differences between soil moisture values at low-slope and high-slope areas over grasslands only, and only in a small region over the UK. The observed relation is relatively low (RЁЕ̨2 = 0.21) and the authors conclude "[...] a topographic signal can be seen in high resolution remotely sensed surface soil moisture data [...]. Unfortunately this signal is relatively weak." Moreover, Mason et al. (2016) uses 1 km SAR data for which topographic corrections are applied in the pre-processing, which likely induces a sensitivity of the measurements to topography parameters. These topography corrections are usually not applied to coarse resolution measurements such as the sensors used within the ESA CCI SM, because topographic effects average out at these scales.

AUTHORS RESPONSE: We agree in that more references could improve the main hypothesis driving this effort, so we will include a more detailed background of the main principles driving this approach in a revised version of our work. We argue that new alternative approaches are needed to provide different downscaling soil moisture outputs to compare data-model agreements and advance science.

We also argue that satellite-derived soil moisture data is retrieved by a direct measurement of the dielectric constant of soils representing specific vegetation types and climate conditions (within each pixel) that are also influenced by topographic patterns. We highlight that there is a high correlation (>0.8 in all our model/years) between topographic patterns and the ESA-CCI soil moisture product, which provides evidence that topography has information that could be used for downscaling soil moisture. This argument could be revised to explain how our machine learning approach can identify non-linearities between satellite-derived soil moisture and topographic variables.

We highlight that our main purpose was to generate a soil moisture downscaled product that is independent of climate or vegetation variables and that provides cross validated hypothesis of a more local (nearly 50% higher) spatial resolution compared with the original satellite soil moisture signal. Finally, we clarify that the use of a soil moisture product where vegetation is not used as predictor (i.e., vegetation variables are independent) could be relevant for decreasing spurious correlations in the further use of soil moisture data (in Earth models and other ecological or geo-scientific analysis).

REVIEWER COMMENT: The presented paper itself also does not analyze the predictive power of the used topographic indices for soil moisture (e.g., the godness-of-fit for the obtained regression, variable importance, etc.). Hence, there is no evidence supporting the reliability of topographic indices as predictor for soil moisture, especially on a global scale.

AUTHORS RESPONSE: We highlight that we have provided information about the predictive power of our models. In Table 1 we report indicators of accuracy and predictive power including the correlation between observed and predicted, the root mean squared error, the number of pixels with soil moisture data available for each year and the best parameters (kernel and k) for each kknn model/year. In our reanalysis the predictive power of our modeling approach increased thanks to the use of a more extensive set of prediction factors. We have also bootstrapped our statistical models and now they provide uncertainty estimates associated with the number of available data

for modeling soil moisture patterns.

In a revised version we can include a variable importance analysis (by permutation) to support/generate new hypothesis about the spatial variability of soil moisture, in relation to land surface characteristics represented by multiple sources of environmental information. We can also provide a bootstrapping approach to estimate uncertainty associated with the number of available data for modeling soil moisture patterns.

REVIEWER COMMENT: Even more doubtable is the assumption that the developed regression function can be used to extrapolate soil moisture to regions not covered by the ESA CCI SM, which are mainly the arctic ice sheet and tropical forests. Tropical rainforests, for example, have a quite unique moisture regime that is expected to be largely rainfall dependent. It is very questionable to use a soil moisture - topography relation that is trained over non-tropical regions to predict soil moisture there. Moreover, no in situ measurements are available in these regions to verify the validity of these predictions.

AUTHORS RESPONSE: We agree that soil moisture estimates should be removed from the arctic ice sheet in a revised version.

However, we believe that our approach could be used for predicting soil moisture across tropical forests. We highlight that there are pixels with satellite soil moisture values across the tropical rain forests of the world, including the Amazon and Congo regions that could be used for prediction within our framework (Fig 1).

In a revised version we will update the soil moisture covariate space with information on ecological domains (FAO, 2010) and bioclimatic features (Fick and Hijmans, 2017), soil type variability (Wieder et al., 2014) and geo-spatial information (Møller, et al., 2019) of available data/pixels in order to improve our soil moisture prediction capacity. We believe that comparing and testing multiple modeling approaches across these areas is needed for improving prediction capabilities for downscaling soil moisture. We also believe that working towards data sharing and open source practices will increase the

users of soil moisture information, increasing the possibilities to solve uncertainties and explain discrepancies between multiple soil moisture products.

REVIEWER COMMENT: Also, the presented validation does not support the conclusions. First, the statement (L234) "In all cases, the evaluation statistics are equal or better for the downscaled soil moisture predictions based on digital terrain analysis (Table 3) than the original ESA-CCI soil moisture product (Table 2)" is wrong. In fact, results are quite balanced, sometimes the downscaled product is "better", sometimes the original is "better", but most likely results are not actually distinguishable within reasonable confidence limits (which should be estimated). The authors do indeed acknowledge (L252): "The downscaled predictions based on digital terrain analysis are not significantly different compared with the ESA-CCI soil moisture product [...]", but the subsequent conclusions are not supported. Specifically, "[...] but they provide (1) gap free soil moisture-related information ": while they are provided, there is no evidence that they are of any reasonable accuracy (for the earlier discussed regions), and "(2) higher resolution (from 27 to 15 km grids)": This is a mix-up of resolution and sampling.

AUTHORS RESPONSE: We highlight that this is an empirical model (i.e., a statistical learning process driven by a machine learning algorithm) and we used a cross validation to show the accuracy in the change of resolution between the original satellite estimate and the prediction; thus it is not only a spatial re-sampling exercise.

We agree with the reviewer in that the results of the validation against filed information are balanced. There are no significant differences in the relationship between the downscaled estimate and the original product, which demonstrates that the downscaled product is preserving the reliability of the ESA-CCI soil moisture product. This is expected as the information that was used for building predictions of soil moisture were the pixel/values of the ESA-CCI soil moisture products. In the revised version of our data paper we will improve the narrative and the explanation on this comparison to avoid misunderstandings.

REVIEWER COMMENT: Improved resolution would imply that there is different / more information in the downscaled product, but the indistinguishability of performance metrics (see above) suggests that this is not the case.

AUTHORS RESPONSE: We argue that the 'indistinguishability of performance metrics' is a proof of the reliability of our prediction framework. Please note that we are using a data driven model were the selection of best parameters (e.g., distances, kernels, neighbors, predictors) is made by cross validation. Cross validation allows to generate unbiased residuals and identify the linear relationship between observed and predicted data. Therefore, it is expected that our predictions to maintain the general pattern (the similar mean and standard deviation) to the general pattern of the ESA-CCI product, but revealing a physiography modulated (not random) soil moisture spatial pattern across finer grids. The soil moisture variability estimated within each 15km pixel is meant to maintain the numerical integrity of the ESA-CCI product, as we also recognize that 15km grids is still too coarse to identify local controls of soil moisture.

In the revised version of our paper, we propose to increase the reproducibility of the soil moisture prediction framework across multiple scales (country-to-global) and highlight the gain of information between the original satellite and the downscaled predictions. We believe that the value of our downscaled product will be better recognized when the reader see the detail gained within a smaller region of the world (i.e., a country).

REVIEWER COMMENT: Also the comparison in Figure 6 (b and c) shows that the original and the downscaled products exhibit the exact same behaviour with a slightly lower overall variability in the original product. It is, however, not clear whether this different overall magnitude reflects an actual improvement, because soil moisture varaibility and trends are actually supposed to be different at a point scale and at a satellite scale (see e.g. Famiglietti et al. 2008).

AUTHORS RESPONSE: We clarify that we indeed found significant differences between the point scale (1:1), and the ESA-CCI soil moisture ($\sim$27km) trends (comparing

only the pixels with field stations) (see Figure 6 of submitted manuscript version). We also found that our downscaled predictions (15km) has a better model-data agreement. We clarify that we only predicted at the places with available data from the ISMN. At these places, we found statistically different trends between the ESA-CCI soil moisture product, the downscaled predictions and the ISMN annual averages.

Although these trends are different, they are all negative and significant (as the confidence intervals are not overlapping zero values) in the three datasets. We found that the ISMN is showing the strongest negative trend, followed by the trend of our downscaled predictions and then the ESA-CCI soil moisture product. Thus, the trend reported by our predictions is closer to the trend of field stations compared with the trend from the ESA-CCI.

In a revised version we can include a comparison of the satellite soil moisture mean using both products and different geographical extents in order to enrich the discussion about the scale dependent variance of soil moisture.

REVIEWER COMMENT: Also, the soil moisture mean is supposed to be different at different scales, hence the negative bias between point and satellite measurements cannot be reliably interpreted as error, and a reduction of this bias may as well be a going in the wrong direction with respect to the true areal-average mean. In other words, even though the generated product is sampled on a higher-resolution grid, it can not be concluded that this product contains higher-resolution information. Given the low amount of evidence that topography (alone) is a go od predictor for soil moisture, observed differences may well be a result of the smoothing-nature of the KKNN approach, and any spatial-window resampling approach may lead to a seemingly "higher-resolution" (which is truly only a higher-sampling) product with the same (or even better) performance, but this is not tested.

AUTHORS RESPONSE: This is an important comment and we argue that is a science frontier that still needs active research. In a revised version we could elaborate about

how different publications address the scale-variance of soil moisture.

We have done a reanalysis of our approach using the recently released ESA-CCI soil moisture version 4.5 and we can confirm a large correlation between the ESA-CCI and our soil moisture predictions (>0.92) at the global scale. Continental to global scales may be consistent in the overall range of values and spatial patterns, however smaller regions may highlight higher differences (Fig 2).

REVIEWER COMMENT:Therefore, I recommend to reject this publication. However, I do believe that topography may well be an important complementary predictor for soil moisture at higher-resolution when combined with other dominant factors. I therefore encourage the authors to pursue this approach addressing the concerns outlined above.

AUTHORS RESPONSE: We are confident that we can address all concerns outlined by the reviewer in a revised version of this manuscript. We believe most of the concerns could be addressed by editing the text to improve clarification and performing new analyses about variable performance, using the recently released version 4.5 of the ESA-CCI, and demonstrate the applicability of our methods at higher temporal resolutions (months) and across smaller areas and spatial extents. Please note that our previous work has demonstrated the effectiveness of our approach at the continental scale of CONUS using 1km grids (Guevara and Vargas, 2019).

References:

Fick, S. E. and Hijmans, R. J.: WorldClim 2: new 1-km spatial resolution climate surfaces for global land area, International Journal of Climatology, 37(12), 4302–4315, doi:10.1002/joc.5086, 2017.

FAO. Global Ecological Zones for FAO Forest Reporting: 2010 Update; FAO: Rome, Italy, 2012.

Guevara, M. and Vargas, R.: Downscaling satellite soil moisture using geomor-

phometry and machine learning, edited by B. Poulter, PLOS ONE, 14(9), e0219639, doi:10.1371/journal.pone.0219639, 2019.

Mascarro, G., Ko, A. and Vivoni, E. R.: Closing the Loop of Satellite Soil Moisture Estimation via Scale Invariance of Hydrologic Simulations, Scientific Reports, 9(1), doi:10.1038/s41598-019-52650-3, 2019.

Møller, A. B., Beucher, A. M., Pouladi, N. and Greve, M. H.: Oblique geographic coordinates as covariates for digital soil mapping, , doi:10.5194/soil-2019-83, 2019.

Wieder, W.R., J. Boehnert, G.B. Bonan, and M. Langseth. 2014. Regridded Harmonized World Soil Database v1.2. Data set. Available on-line [http://daac.ornl.gov] from Oak Ridge National Laboratory Distributed Active Archive Center, Oak Ridge, Tennessee, USA. http://dx.doi.org/10.3334/ORNLDAAC/1247.

[Figure]

**Fig. 1.** Soil moisture available data in Tropical Rain Forests (ESA-CCI 4.5, 2018) (a). Soil moisture prediction (b) and soil moisture prediction variance (c).

[Figure]

**Fig. 2.** Example of soil moisture predictions for 2018 (i), variances for 2018 (ii), the training data image for 2018 (iii) and their statistical distribution ('boxplots').

---

## Short Comment (SC2) · 31 Dec 2019

REVIEWER COMMENT: OVERVIEW The study has developed a gap-filled, and downscaled with topography-derived information, long-term annual (15 km) global soil moisture dataset based on the ESA CCI satellite soil moisture products. The assessment of the dataset with respect to in situ observations has been carried out through annual comparisons as well as in terms of long-term trends (1991-2016).

REVIEWER COMMENT: GENERAL COMMENTS The paper is mostly well written and clear. The topic of the paper is interesting for the readership of Earth System Science Data as a global scale gap-filled annual soil moisture dataset is surely useful for many applications. However, I believe the paper needs major changes before the publication

as several parts are not properly described and other sections need to be improved or summarized. I have listed below my comments with the indication of their relevance.

AUTHORS RESPONSE: We appreciate the reviewer comments. We can clarify and improve the narrative of our methods in a revised version.

REVIEWER COMMENT:1) MAJOR: The factors used in the downscaling and gap-filling algorithm should be described in details. Two figures (Figures 2 and 3) are not mentioned in the text. It seems a part is missing. The reader needs to know the details on the methodology employed and which factors have been found to be more important. Which Digital Elevation Model is used?

AUTHORS RESPONSE: We fully agree with this comment and will include this information in the revised version of our manuscript.

REVIEWER COMMENT: Additionally, other static factors such as vegetation and soil types are not considered. Why?

AUTHORS RESPONSE: Our initial objective was to test the predictive capacity of topographic terrain parameters derived from a single source of information (elevation), considering that a satellite soil moisture pixel is representative of soil moisture of the spatial configuration of climate and ecological conditions (within each pixel) for a specific period of time.

In a revised version we will update our soil moisture prediction framework and include (as prediction factors) bioclimatic and soil classes static information. In a new version of our paper we will compare the predictive capacity of topographic patterns in relation to these bioclimatic and soil classes.

REVIEWER COMMENT: The discussion on the approach employed for downscaling and gap-filling needs to be included in the paper. Why do the authors select such approach?

AUTHORS RESPONSE: We will include more information on model selection in the

revised version of our manuscript. Briefly, we used a kknn algorithm because it is fast and yielded reasonable good accuracy (r=>0.8 in all models) in comparison with other modeling approaches that are computationally more expensive. Algorithms such as Random Forests or Support Vector Machines are examples of conventional machine learning methods that can be used as regressors for satellite soil moisture gridded surfaces. We decided to use kknn to generate a baseline of predictions that can be reproduced in hours (annual 1991-2018) using conventional laptops (i.e., 6GB of RAM) in order to increase the users of soil moisture information.

In addition in a revised version we will include a bootstrapping technique applied to the kknn prediction framework to account for the sensitivity of the model to variations in datasets. Also, detailed information on variable importance for kknn models will be included in the revised version as well as an appendix with a performance comparison against other prediction algorithms.

REVIEWER COMMENT: 2) MAJOR: I have found particularly challenging performing gap-filling over dense forest regions in the Amazon and in Congo. Satellite soil moisture data cannot be used in such regions due to dense forest that mask the soil moisture signal. How is it possible to extend the signal there only based on topography?

AUTHORS RESPONSE: There are sparse pixels with soil moisture data across specific areas of the tropical rain forest such as the Amazon or Congo that are useful for modeling and calibrating soil moisture predictions using our approach (Fig 1). In a revised version we will include new prediction factors to account for ecological and climate variability (e.g., bioclimatic variables).

Soil type information (e.g., harmonized world soil database) could also be included to account for the capacity of soil to retain water across these areas with low availability of soil moisture data in the ESA-CCI soil moisture product. We will include a comparison of model performance using multiple combinations of prediction factors. We expect that with a better description of the methods and this revised approach we will provide
a better explanation of our alternative approach to downscale soil moisture across the world.

REVIEWER COMMENT: I would suggest to perform a more detailed validation in these areas. I strongly suggest to perform a comparison with modelled datasets (e.g., ERA5 soil moisture) to have an assessment of the performance over dense vegetated areas. A similar comment can be done for high latitude areas in which frozen soils and snow completely mask the soil moisture signal. Please perform a detailed validation over these areas, too.

AUTHORS RESPONSE: We appreciate the comment and we will improve our validation across these areas (e.g., tropical forests or high latitude areas) using in situ records of temperature, precipitation and ecological patterns synthesized in previous work (Bond-Lamberty and Thomson, 2018). For example, previous studies have described the coupling between soil moisture and precipitation (Koster, et al., 2004, McColl et al., 2017).

REVIEWER COMMENT: 3) MAJOR: The trend analysis is very interesting. However, as above, we need more details on how trends are computed. For instance, in situ stations are available only at some points over the Earth, are the same locations used with the satellite-derived datasets? If not, the comparison is wrong.

AUTHORS RESPONSE: We will improve the description of the methods in a revised version of our manuscript. We clarify that the comparison was done using only pixels where there was a spatial match with the sites of the ISMN.

REVIEWER COMMENT: Similarly, in situ stations are not available every year, and for the full year. How are the data aggregated in time and space? These details are needed. I expect the results are strongly impacted to these choices.

AUTHORS RESPONSE: We recognize that there is limited soil moisture field information for validating models and satellite soil moisture estimates across large areas of the
world. We will provide more information about soil moisture (historic and current) data availability in the ISMN and its use for this study. We will also provide more detailed information on how this dataset was aggregated.

REVIEWER COMMENT: 4) MODERATE: The machine learning downscaling approach provides soil moisture data with a resolution higher than the original ESA CCI product. However, I am always doubtful on these downscaling approaches as instead of resolution it should be higher spatial sampling. The higher spatial resolution should be tested, but I am aware it is very hard to do (I have this comment for all downscaling studies).

AUTHORS RESPONSE: We fully agree with the reviewer comment and we recognize that there is a compromise between where and when to sample across scales. We also recognize that all global studies are limited with the available information of global networks and local studies (across multiple ecosystems and regions of the world) are needed to better test downscaling approaches. In a revised version we will include an improved validation including new information data across poorly represented areas.

REVIEWER COMMENT: The authors should demonstrate that the downscaled product is able to reproduce features at higher resolutions with respect to the parent ESA CCI product. It is not done in the paper, that's why I believe higher spatial sampling, and not spatial resolution, is more appropriate.

AUTHORS RESPONSE: We found a larger range of soil moisture predicted values compared with the original ESA-CCI soil moisture product. We found a temporal trend at the places of field stations that is more similar between the field data and our predictions compared with the ESA-CCI soil moisture product. Please note that the main purpose of the model is to reproduce the signal of satellite soil moisture using as reference the relationship that it maintains with topographic data. This is a regression problem were the satellite soil moisture measurements (for a specific time across an area, a pixel under a approximately the same vegetation type or general climate con-

dition) are statistically related to multiple quantitative topography surrogates.

We believe that a spatial resampling (Fig 2a) is just a change of spatial resolution by using simple algorithmic approaches across the orthogonal relationship of the variable itself with the latitude and longitude plane. In contrast, our models (Fig 2b) are replicated and there is a statistical learning process on each iteration for the selecting optimal model parameters and minimizing the prediction error given a specific spatial resolution defined by the topographic prediction factors. We understand that this represents a conceptual and semantic debate and we will elaborate in a revised version.

REVIEWER COMMENT: 5) MODERATE: Several performance scores have been used in the paper. However, I don't think it is necessary to use all of them. The authors should discuss what information each performance score is providing for the assessment of the dataset, not simply to list many numbers. Indeed, Tables 2 and 3 are hard to read and not informative. Please summarize only the more relevant scores in a figure.

AUTHORS RESPONSE: We can summarize the accuracy numbers of these reports and we can improve the discussion around them. We will also include a figure (i.e., from multivariate analysis) showing the main differences in the revised version of our manuscript.

REVIEWER COMMENT: 6) MAJOR: The range of values of ESA CCI soil moisture products has little value, as the satellite products are rescaled to match the range of variability of modelled soil moisture from GLDAS. Therefore, the range of values is that obtained from GLDAS. For the analysis shown in Figure 4, and similarly for the trend analysis, the soil moisture datasets should be rescaled between the minimum and maximum of each time series and expressed as relative soil moisture (between 0 and 1). Then the data should be aggregated and the range of values and the trends can be assessed.

AUTHORS RESPONSE: We can transform the values of the soil moisture maps to

relative soil moisture and repeat the trend analysis and the comparison with field data. We will update trend results in a revised version of our approach.

7) MODERATE: I believe the discussion section must be rewritten. General results are mostly discussed, whereas it should be closely related to the results shown in the paper. I believe it should be shorter and better focused.

AUTHORS RESPONSE: We can improve the narrative of our discussion and main findings.

REVIEWER COMMENT: SPECIFIC COMMENT (L: line or lines) L307: Why the "angle between satellite sensors and the earth surface" is useful for determining soil moisture? It has no sense and I believe it is wrong.

AUTHORS RESPONSE: We meant to say that topography affect the distance between the satellite and the earth surface and therefore It could be correlated with the satellite soil moisture signal (which is a hypothesis proven with the data analyzed in this study).

REVIEWER COMMENT: RECOMMENDATION Based on the above comments, I suggest a major revision before the possible publication on Earth System Science Data.

AUTHORS RESPONSE: We appreciate the comments of the reviewer that will improve the overall revised manuscript.

References:

Bond-Lamberty, B.P., and A.M. Thomson. 2018. A Global Database of Soil Respiration Data, Version 4.0. ORNL DAAC, Oak Ridge, Tennessee, USA.Âăhttps://doi.org/10.3334/ORNLDAAC/1578

McColl, K. A., Alemohammad, S. H., Akbar, R., Konings, A. G., Yueh, S. and Entekhabi, D.: The global distribution and dynamics of surface soil moisture, Nature Geoscience, 10(2), 100–104, doi:10.1038/ngeo2868, 2017.

Koster, R. D.: Regions of Strong Coupling Between Soil Moisture and Precipitation,

Science, 305(5687), 1138–1140, doi:10.1126/science.1100217, 2004.

[Figure]

[Figure]

**Fig. 1.** Training data for 2018 across Tropical Rain Forests (a), the model prediction for 2018 (b) and the prediction variance after multiple realizations for 2018 (c).

[Figure]

**Fig. 2.** Simple spatial resampling (bilinear) applied to the ESA-CCI product from 27 to 5 km grids (a) and our modeling output using 5km grids (b) across France.

---

## Short Comment (SC3) · 31 Dec 2019

REVIEWER COMMENT: General comments: While seeking a higher resolution global soil moisture product is certainly a laudable goal, I find the methods used in this paper lack a credible connection between ground measurements and the remotely sensed data. The authors use machine learning to generate regressions between multiple topographic variables thought to be related to soil moisture that are available at higher resolution with coarse resolution satellite data. It is difficult for the reader to discern if actual new information results from the downscaling because there is not a clear connection made between insitu soil moisture measurements, their physical connection to the chosen topographic variables used, and resultant satellite measurements.

AUTHOR RESPONSE: We will improve the clarity of our methodological approach in a revised version of our manuscript to highlight why topographic terrain parameters are hydrologically meaningful. We would like to clarify that in situ observations are used for validating purposes only. In this study, the main purpose was to compare if the fusion of satellite soil moisture records and elevation-derived terrain parameters (by the means of an empirical modeling approach, not physical) was useful to fill gaps of satellite estimates. The physical connection between soil moisture and terrain attributes is that these attributes regulate the overland flow and the potential solar radiation income. Both process are controlled by topography and therefore should show a direct influence on soil moisture patterns. We will also include more information supporting the use of these hydrologically meaningful terrain parameters for predicting soil moisture patterns, an emergent research opportunity on Geomorphometry.

REVIEWER COMMENT: To do so, requires first demonstrating the rigor of the methodology over a much smaller and better measured area, such as the area of the International Soil Moisture Network.

AUTHOR RESPONSE: Our approach has already been tested across the contiguous United States (Guevara and Vargas, 2019). Thus, this study is an extension of that demonstrated methodology and now is applied to the global scale.

In a revised version of our paper, we will compare and test the accuracy of soil moisture predictions using country specific and global models to demonstrate the consistency of this approach across multiples scales of data availability. Using the North American Soil Moisture Database (Quiring, 2016) across the state of Oklahoma, in the Great Plains of the US (one of the areas with highest in situ soil moisture records for validating purposes, Llamas et al., 2019).

REVIEWER COMMENT: Extending the algorithm to areas that are hydrogeomorphically and climatically distinct from the existing soil moisture measurement networks cannot be result in credible data. I do not recommend publication of this work.

AUTHOR RESPONSE: As explained in previous work (i.e., Gessler et al., 2009, Florin-sky, 2012, MacMillan et al., 2016), terrain parameters (such as those we used for predicting soil moisture) both influence the accumulation of surface geological materials and reflect this spatial distribution. Terrain parameters such as the wetness index or the relative slope position both influence the flow and accumulation of moisture and reflect it. Terrain parameters both influence the spatial patterns of distribution of vegetation/land use and reflect these patterns (MacMillan et al., 2016). Our methods are based on geomorphometry which differs from a purely physical hydrological model. That said the novelty of our study is that it takes the variables to a machine learning approach in the covariate space using as training data the satellite soil moisture pixels in the latest version of the ESA-CCI, which are soil moisture values representative of a mosaic of ecological and environmental conditions (for a specific time) within an area (pixel) of around 27km of spatial resolution. The strength of a machine learning model as kknn is to find non-linear relationships in complex spaces to recreate patterns.

These patterns are dependent of training data and there are sparse pixels with valid soil moisture data across dense vegetation areas or high latitudes that are useful to support the reliability of our soil moisture predictions (Fig 1). Replicating our models can give us information about modeling variance across areas with sparse training data . We re-analyzed our datasets and updated soil moisture predictions based on the most recent version of the ESA-CCI soil moisture product (to 2018) and included multiple sources of prediction factors (e.g., bioclimatic, soil type information) in order to maximize the reliability of our prediction framework and highlight the importance of terrain parameters modeling soil moisture.

In a revised version we will use data from the ISMN and other published values of soil moisture to expand our validation dataset. These new validation analyses can be performed across different biomes rather than at the global scale.

REVIEWER COMMENT: Major comment: Much more information is needed about the regression methods and parameter selection.

AUTHOR RESPONSE: We will improve the narrative of the kernel based machine learning algorithm we used to perform regression. The main parameters of this method (kknn) are the kernel type (used to convert distances between neighbors points in the statistical space, to weights) and the k parameter (the number of neighbors used to calculate a weighted average in regression). These parameters are selected in our modeling approach automatically using repeated cross validation (Table 1 of submitted paper). The terrain parameters derived by the means of Geomorphometry from the digital elevation model are another component of our modeling framework, used prediction factors (soil moisture covariates) in the kknn approach.

REVIEWER COMMENT: It would be helpful to discuss what each parameter is mathematically and why it can be useful in a soil moisture prediction context.

AUTHOR RESPONSE: We will explain with more detail why the terrain parameters derived from elevation data using Geomorphometry are hydrologically meaningful, and will include also mathematical formulations to identify the main impacts of topography in soil moisture. Detailed information on these terrain parameters is reported in our previous study across the conterminous United States (see table: https://doi.org/10.1371/journal.pone.0219639.s007).

REVIEWER COMMENT: Can it be demonstrated over smaller areas that there is a valid argument for using these variables, some or all of them.

AUTHOR RESPONSE: We tested our approach across continental and state scales and found consistent results. Across the conterminous US, we recently found an increase of nearly 25% of accuracy validating our predictions against the North American Soil Moisture Database (https://doi.org/10.1371/journal.pone.0219639.g005).

REVIEWER COMMENT: Explain in detail the cross validation process used and why.

AUTHOR RESPONSE: We will include more information and references about cross validation strategies for re-sampling and bootstrapping prediction models. Cross validation is a family of re-sampling techniques that are used to analyze the sensitivity of models (in this case) to variations of available data for training purposes. Multiple models are generated using multiple proportions of available data and validating with subsets of information that are leaved out of these models. This process is repeated several times until capturing the magnitude of variance associated with the models based on the data subsets.

REVIEWER COMMENT: It is unclear if there is a single model developed and applied to all years or separate models for each year. Are all topographic parameters used in the model in all years? What is their weighting?

AUTHOR RESPONSE: We generate and cross validate a model for each year of available data. Table 1 shows the results of cross validation and available data for each model/year. All topographic parameters are used on each model/year. During the cross validation, kknn uses a kernel form to convert distances in weights that are used for prediction. These distances are different from one place to another as the pattern recognition includes all the points and their k neighbors. Using the independent residuals of each model realization during the cross validation strategy, the optimal weights are selected using as indicators the root mean squared error and the correlation between observer and predicted (also included in Table 1 of submitted manuscript). After all model realizations including all possible kernel combinations (e..g, "rectangular", "triangular", "epanechnikov","gaussian", "rank", "optimal") and increasing the number of neighbors (k) systematically (e.g., from 2 to 25, See Table 1) we find for each year the optimal weights maximizing the correlation between observed and predicted and minimizing the root mean squared error. In our reanalysis, we have included a novel variable importance analysis (by permutation) for kknn, that allows to identify which are the most important variables in the overall prediction using this kernel based algorithm.

REVIEWER COMMENT: Major comment: Extending the results to areas without soil moisture measurements (whether in situ or remotely sensed values) without any validation is not an improvement over current spatial data. It is highly suspect and could

lead to inappropriate applications using what is in effect non-data.

AUTHOR RESPONSE: We agree that any modeling output should be interpreted and used carefully as it is not a direct measurement. However, we performed a robust cross validation strategy including data available in the ESA-CCI representing all environmental conditions and provide information about data-model agreement. Then we validate with field data only at the places (pixels) containing in situ information in the International Soil Moisture Network. It is clear that contiguous information on soil moisture is challenging to obtain across large areas of the world such as polar or tropical rain forest conditions. However there are limited and sparse but representative satellite soil moisture records across these Polar (Fig 1a) or tropical areas (Fig 1b) that can be used to improve the spatial representation of global soil moisture patterns.

We will include in our reanalysis more datasets (Fig 1c) and more validation information to enrich the discussion about the reliability of our prediction framework for soil moisture (e.g., Bond-Lamberty and Thomson). Please note that we are following a conceptual and data-driven framework assuming that comparing and testing multiple modeling approaches is still required to better understand the implications of soil moisture on ecological patterns across areas where no soil moisture information is available. Thus, our results provide some evidence that our soil moisture product is useful to better understand ecological patterns in places with low availability of satellite soil moisture estimates.

REVIEWER COMMENT: Moderate comment: I found the paper very repetitive and in need of detailed editing.

AUTHOR RESPONSE: We will revise narrative and overall organization of the manuscript.

References

Bond-Lamberty, B.P., and A.M. Thomson.    A Global Database of Soil Respiration Data, Version 4.0. ORNL DAAC, Oak Ridge, Tennessee, USA. https://doi.org/10.3334/ORNLDAAC/1578, 2018 Florinsky, I. V.: Influence of Topography on Soil Properties, Digital Terrain Analysis in Soil Science and Geology, 145–149, doi:10.1016/b978-0-12-385036-2.00008-0, 2012.

Gessler, P., et al., Chapter 28 The Future of Geomorphometry, Developments in Soil Science, 637–652, doi:10.1016/s0166-2481(08)00028-7, 2009.

Guevara, M. and Vargas, R.: Downscaling satellite soil moisture using geomorphometry and machine learning, edited by B. Poulter, PLOS ONE, 14(9), e0219639, doi:10.1371/journal.pone.0219639, 2019.

Llamas, R. M., Guevara, M., Rorabaugh, D., Taufer, M. and Vargas, R.: Spatial Gap-Filling of ESA CCI Satellite-Derived Soil Moisture Based on Linear Geostatistics, , doi:10.20944/preprints201909.0126.v1, 2019.

MacMillan et al., A hydrologically explicit, spatially exact, classification of landforms for Canada at 1:500,000 scale., EGUGA, EPSC2016-13103 https://ui.adsabs.harvard.edu/abs/2016EGUGA..1813103M/abstract (Accessed 23 December 2019), 2016.

Quiring, Steven M., Trent W. Ford, Jessica K. Wang, Angela Khong, Elizabeth Harris, Terra Lindgren, Daniel W. Goldberg, and Zhongxia Li. 2015. "The North American Soil Moisture Database: Development and Applications." Bulletin of the American Meteorological Society, November. doi:10.1175/BAMS-D-13-00263.1.

[Figure]

[Figure]

**Fig. 1.** Available information in the ESA-CCI (4.5, 2018) across Polar (a) and dense vegetated areas and tropical rain forests (b). In situ dataset that we will use to improve our validation effort (c).

---

## Author Comment (AC1) · 29 Jan 2020

**"Gap-Free Global Annual Soil Moisture: 15km Grids for 1991–2016" by Mario Guevara et al. Responses to anonymous reviewer #1**

REVIEWER COMMENT: This paper presents a 15 km annual-average soil moisture product that is generated by machine-learning the relation between 0.25 degree ESA CCI soil moisture estimates and topographic indices derived from a higher-resolution DEM.
I have several major concerns regarding the hypothesis/assumptions on which the methodology is based as well as the employed validation methodology, and consequently also the conclusions drawn from the presented analysis:

AUTHORS RESPONSE: We appreciate the reviewer comments as they provide valuable feedback to increase the impact of our work. We have revised our work and provided new analyzes to clarify our framework and improve the validation methodology. We have updated our work with new datasets (e.g., eco-climatic and soil type classes) and updated our results with the recently released version of the ESA-CCI soil moisture product (4.5, up to 2018). These results have been uploaded as an updated version of our data repository. We clarify that the original data-product has not substantially changed, but we have added new model outputs and further analyzes to address the concerns from the reviewers.
        To increase reproducibility and transparency of our prediction framework we compiled a set of R functions (link available with the new version of the manuscript). This compilation is useful to predict soil moisture based on remote sensing and machine learning, from the global to country-specific scales. Furthermore, we now include a bootstrapping approach given a user defined sample size (if not specified, it will use 1/3 of available data/pixels for each year) to analyze the variance of model predictions as a function of variations in training data. Thus, in the revised version of the manuscript we report a spatial explicit metric of model-based uncertainty for the fusion of soil moisture satellite estimates and topographic constraints. We believe that these advances (including new model estimates and uncertainty estimates) constitute a substantial improvement from the previous version of the manuscript.

REVIEWER COMMENT: The methodology is based on the hypothesis that topography is a main driving factor for soil moisture patterns. However, the reference used to support this claim (Mason et al., 2016) presents only a very local analysis of differences between soil moisture values at low-slope and high-slope areas over grasslands only, and only in a small region over the UK. The observed relation is relatively low ($R^2 = 0.21$) and the authors conclude "[...] a topographic signal can be seen in high resolution remotely sensed surface soil moisture data [...]. Unfortunately this signal is relatively weak." Moreover, Mason et al. (2016) uses 1 km SAR data for which topographic corrections are applied in the pre-processing, which likely induces a sensitivity of the measurements to topography parameters. These topography corrections are usually not applied to coarse resolution measurements such as the sensors used within the ESA CCI SM, because topographic effects average out at these scales.

AUTHORS RESPONSE: We have improved the introduction with more references and examples. We also clarified the main hypothesis driving this effort (i.e., that topographic terrain parameters are good predictors for downscaling satellite-derived soil moisture).

We argue that satellite-derived soil moisture data is retrieved by a direct measurement of the dielectric constant of soils that is representing specific vegetation types and climate conditions (within each pixel) that are intrinsically influenced by topographic patterns. We highlight that there is a high correlation (Table 1) between topographic patterns and the ESA-CCI soil moisture product, which provides evidence that topography has information that could be used for downscaling satellite soil moisture. Our machine learning approach is able to reproduce non-linearities between satellite-derived soil moisture and topographic variables, and generate predictions of soil moisture across higher resolution spatial grids.

In the new version we compared the prediction capacity of multiple prediction factors of soil moisture (topographic terrain parameters, bioclimatic features and soil types) to identify the value of terrain parameters predicting soil moisture. We highlight that our main purpose was to generate a soil moisture downscaled product that is independent of climate or vegetation variables and that provides cross validated hypothesis of a more local (nearly 50% improvement) spatial resolution compared with the original satellite soil moisture signal. In the updated version of our soil moisture datasets, we explicitly account for the spatial relationships of soil moisture available data and the geographical space following a digital soil mapping spatial prediction framework (McBratney et al., 2003, Hengl et al., 2018, Møller et al., 2019) to increase the reliability of our predictions. Finally, we clarify that the use of a soil moisture product where vegetation or climate data are not used as predictors could be relevant for decreasing potential spurious correlations in the further use of our soil moisture data-product.

REVIEWER COMMENT: The presented paper itself also does not analyze the predictive power of the used topographic indices for soil moisture (e.g., the godness-of-fit for the obtained regression, variable importance, etc.). Hence, there is no evidence supporting the reliability of topographic indices as predictor for soil moisture, especially on a global scale.

AUTHORS RESPONSE: We highlight that we have provided information about the predictive power of our models. In Table 1 we reported indicators of accuracy and predictive power including the correlation between observed and predicted, the root mean squared error, the number of pixels with soil moisture data available for each year and the best parameters (kernel and k) for each kknn model/year. In the revised version, the predictive power of our modeling approach increased thanks to the use of a more extensive set of prediction factors. We have also bootstrapped our statistical models and now they provide uncertainty estimates associated with the number of available data for modeling soil moisture patterns.

In the revised version we include a simple variable importance analysis (by permutation) to support/generate new hypothesis about the spatial variability of soil moisture, in relation to land surface characteristics represented by multiple sources of environmental information. In addition, we provide a bootstrapping approach to estimate uncertainty associated with the number of available data and predictors for modeling soil moisture patterns.

REVIEWER COMMENT: Even more doubtable is the assumption that the developed regression function can be used to extrapolate soil moisture to regions not covered by the ESA CCI SM, which are mainly the arctic ice sheet and tropical forests. Tropical rainforests, for example, have a quite unique moisture regime that is expected to be largely rainfall dependent. It is very questionable to use a soil moisture - topography relation that is trained over non-tropical regions to predict soil moisture there. Moreover, no in situ measurements are available in these regions to
       verify the validity of these predictions.

AUTHORS RESPONSE: We agree that soil moisture estimates across the arctic ice sheet are not informative; therefore, these areas have been removed from our analysis and the model outputs in the revised version.

We recognize that in-situ global soil moisture field datasets are limited for validating our soil moisture predictions across tropical areas, but we have done efforts to bring more available in-situ measurements in this revised version. That said, we demonstrate the reliability of our predictions by comparing the original satellite and our predictions against in-situ records of rainfall patterns across the tropical areas of the world (171 sites across tropical areas of the world for the years 2008 to 2018). We now report and improvement with our predictions and in-situ precipitation records across multiple biomes of the world (e.g., r=031 to r=0.38 across tropical biomes and r=40 to r=51 across temperate biomes) using information from previous studies.

         We argue that our approach can be used for predicting soil moisture (and associated uncertainty) across tropical forests. We highlight that our framework is an empirical approach using machine learning algorithms that are able to reproduce patterns extracted from a multivariate space. We highlight that there are pixels with satellite soil moisture values across the tropical rain forests of the world, including the Amazon and Congo regions that could be used for prediction within our framework (Figure R1_1).

         In the revised version we compare a model using terrain parameters as prediction factors of soil moisture and a model including information on bioclimatic features (FAO, 2010, Fick and Hijmans, 2017) and soil type variability (Wieder et al., 2014). We use geo-spatial information of available data/pixels in order to explicitly account for soil moisture spatial relationships between soil moisture available values for training the models. Finally, we report uncertainty estimates to provide spatial information about model performance and to identify regions with high model uncertainty. We belive that this new addition (reporting uncertainty) is a much needed effort in local-to-global predictions.

[Figure]

Figure R1_1Soil moisture across Tropical Rain Forests of the world based on the data available in the
ESA-CCI soil moisture product (4.5) for the year 2018 (a). We show the soil moisture prediction (b),
the soil moisture prediction variance using only the data available for Tropical Rain Forests (c). Note
that the correlation between observed and predicted decreased to 0.62, most likely due to the limited
information for modeling these ecosystems, however the root mean squared error is comparable with a
model using all global data (e.g., <0.04).

In the revised version we compare a model using terrain parameters as prediction factors of soil
moisture and a model including information on bioclimatic features (FAO, 2010, Fick and Hijmans,
2017) and soil type variability (Wieder et al., 2014). We use in our modeling approach geo-spatial information of available data/pixels in order to explicitly account for soil moisture spatial relationships between soil moisture available values for training the models.

REVIEWER COMMENT: Also, the presented validation does not support the conclusions. First, the statement (L234) "In all cases, the evaluation statistics are equal or better for the downscaled
soil moisture predictions based on digital terrain analysis (Table 3) than the original ESA-CCI soil moisture product (Table 2)" is wrong. In fact, results are quite balanced, sometimes the downscaled product is "better", sometimes the original is "better", but
most likely results are not actually distinguishable within reasonable confidence limits (which should be estimated). The authors do indeed acknowledge (L252): "The downscaled predictions based on digital terrain analysis are not significantly different compared with the ESA-CCI soil moisture product
[...]", but the subsequent conclusions are not supported. Specifically, "[...] but they provide (1) gap free soil moisture-related information ": while they are provided, there is no evidence that they are of any reasonable accuracy (for the earlier discussed regions), and "(2) higher resolution (from 27 to
15 km grids)": This is a mix-up of resolution and sampling.

AUTHORS RESPONSE: We highlight that this is an empirical model (i.e., a statistical learning
process driven by a machine learning algorithm) and we used a cross validation to show the correlation of the change of resolution between the original satellite estimate and the prediction; thus it is not only a spatial re-sampling exercise. It is an empirical model that relies on satellite soil moisture data and its
spatial relationships with topographic data. Then, independent in-situ data is used to validate the model output.
The reviewer is correct to point that the validation against filed information are balanced
between the CCI-ESA product and our downscaled product. There are no significant differences in the relationship between the downscaled estimate and the original product; therefore, this demonstrates that the downscaled product is preserving the statistical properties of the ESA-CCI soil moisture product.
This is expected as the information that was used for building predictions of soil moisture were the pixel/values of the ESA-CCI soil moisture product without adding any in-situ information (again, the in-situ data was only used for independent validation). In the revised version we have improved the
narrative and the explanation on this comparison for clarity.
Our results comparing the predictive capacity of terrain parameters on soil moisture (in relation to bioclimatic and soil type features), suggest that terrain parameters are useful to fill-gaps in the ESA-
CCI and maintain its accuracy against field observations. We now include a full model (including bioclimatic and soil type information) that suggests a slightly improvement (but not significant) in all model evaluation metrics (Figure R1_2). However, we highlight that our main purpose was to generate
a parsimonious product independent of climate, biological and other sources of information with the main motivation of minimizing spurious correlations in the further use of soil moisture data (in Earth models and other ecological or geo-scientific analysis). Therefore, we conclude that a parsimonious
model based only on terrain parameters performs similarly to a more complex model for predicting soil moisture at 15Km resolution.

REVIEWER COMMENT: Improved resolution would imply that there is different / more information in the downscaled product, but the indistinguishability of performance metrics (see above) suggests that
this is not the case.

AUTHORS RESPONSE: We argue that the 'indistinguishability of performance metrics' is a proof of the reliability of our prediction framework. Please note that we are using a data driven model were the selection of best parameters (e.g., distances, kernels, neighbors, predictors) is made by cross validation. Cross validation allows to generate unbiased residuals and identify the linear relationship between observed and predicted data. Therefore, it is expected that our predictions maintain the general pattern (i.e., the similar mean and standard deviation) of the ESA-CCI product (Figure R1_2). No significant differences were observed between a model based on terrain parameters or the full model (using bioclimatic and soil type classes), but both of these models were better correlated against field data.

[Figure]

Figure R1_2 Evaluation of soil moisture predictions based on quantiles. The relationship between the ESA-CCI and the ISMN in an annual basis (a). We show the relationship between the ISMN field soil moisture and our predictions based on terrain parameters (b) in relation with a model using bioclimatic and soil type classes as prediction factors (c). Blue line is a perfect model. Blue histogram is from training data and gray histogram are from model predictions.

The soil moisture variability estimated within each 15km pixel is meant to maintain the numerical integrity of the ESA-CCI product at the global scale. We highlight that the gain of information between the original satellite and the downscaled predictions when visualizing soil moisture predictions across smaller areas (i.e., countries). We believe that the value of our downscaled product will be better recognized when the user sees the detail gained within a smaller region of the world (i.e., a country) (Figure R1_3).

[Figure]

Figure R1_3 Example of soil moisture predictions based on the extent of countries with different sizes
(e.g., Canada, Australia and Mexico) to show that there is a consistent increase in the range of soil
moisture values in the predicted soil moisture maps of 15km grids compared with the original satellite estimate (~27km grids). The first column of maps contains the predictions, the second column the
prediction variances and the third column the training data for each country specific model. The last
column shows boxplots of the three maps for each country (note that in all cases our predictions reveal a larger range of values compared with the original estimates).

REVIEWER COMMENT: Also the comparison in Figure 6 (b and c) shows that the original and the
downscaled products exhibit the exact same behaviour with a slightly lower overall variability in the original product. It is, however, not clear whether this different overall magnitude reflects an actual
improvement, because soil moisture varaibility and trends are actually supposed to be different at a
point scale and at a satellite scale (see e.g. Famiglietti et al. 2008).

AUTHORS RESPONSE: We would like to highlight that one clear improvement of our soil moisture
predictions is that the soil moisture trend reported by our predictions is closer to the trend detected from soil moisture field stations, when compared with the trend from the ESA-CCI (Figure 6 of
submitted manuscript version). We clarify that this comparison is made only with information at the
places of field stations and associated pixels in the ESA-CCI and our downscaled product.

Although these trends are different, they are all negative and significant (as the confidence
intervals are not overlapping zero values) in the three datasets. We found that the ISMN is shows the
strongest negative trend, followed by the trend of our downscaled predictions and then the ESA-CCI

soil moisture product.
In terms of spatial variability, the scale dependent variance of soil moisture analysis by
Famiglietti (et al. 2008) is focused on shorter spatial scales, but we argue that there is no clear evidence of significant differences in soil moisture at the global scale when comparing the ESA-CCI and our downscaled product. Famiglietti (et al. 2008) provide evidence of scale dependent variances of soil moisture across shorter distances and relatively smaller spatial extents compared with our global effort. Recent efforts also have described scale invariant properties of soil moisture across scales (Mascaro and Vivoni, 2019). Thus, multiscale predictions of soil moisture using multiple modeling approaches could be useful to strive to overcome the main limitations (i.e., spatial gaps) of current soil moisture spatial information and solve the multiscale variability of soil moisture patterns from the plot to the country and global scales. This important discussion is now included in a revised version of the manuscript.

REVIEWER COMMENT: Also, the soil moisture mean is supposed to be different at different scales, hence the negative bias between point and satellite measurements cannot be reliably interpreted as error, and a reduction of this bias may as well be a going in the wrong direction with respect to the true areal-average mean. In other words, even though the generated product is sampled on a higher-resolution grid, it can not be concluded that this product contains higher-resolution information. Given the low amount of evidence that topography (alone) is a good predictor for soil moisture, observed differences may well be a result of the smoothing-nature of the KKNN approach, and any spatial-window resampling approach may lead to a seemingly "higher-resolution" (which is truly only a higher-sampling) product with the same (or even better) performance, but this is not tested.

AUTHORS RESPONSE: This is an important comment and we argue that is a science frontier that still needs active research. In the revised version we discuss the challenge of representing the scale-variance of soil moisture. We highlight that our analyzes between datasets have been done at the global scale and therefore any potential regional or country scale-variance are not included, but could be an important use of the dataset as a follow-up study.
To address the reviewer's comments we have done a reanalysis of our approach using the recently released ESA-CCI soil moisture version 4.5 and we can confirm a large correlation between the ESA-CCI and our soil moisture predictions (>0.92) at the global scale. Continental to global scales may be consistent in the overall range of values and spatial patterns, however smaller regions may highlight potential larger differences (Figure R1_3).

REVIEWER COMMENT: Therefore, I recommend to reject this publication. However, I do believe that topography may well be an imprtant complementary predictor for soil moisture at higher-resolution when combined with other dominant factors. I therefore encourage the authors to pursue this approach addressing the concerns outlined above.

AUTHORS RESPONSE: We believe that our revised version addresses most of the concerns of this reviewer. We have included new model outputs (i.e., full model including soil and bioclimatic factors), developed uncertainty estimates, and revised the analyses to improve clarity. We now clarify several sections of the main text, use the recently released version 4.5 of the ESA-CCI, and demonstrate the applicability of our methods across smaller areas and spatial extents. Please note that our previous work has demonstrated the effectiveness of our approach at the continental scale of CONUS using 1km grids (Guevara and Vargas, 2019).

References:

Bond-Lamberty, B.P., and A.M. Thomson. 2018. A Global Database of Soil Respiration Data, Version 4.0. ORNL DAAC, Oak Ridge, Tennessee, USA.Âahttps://doi.org/10.3334/ORNLDAAC/1578

Fick, S. E. and Hijmans, R. J.: WorldClim 2: new 1-km spatial resolution climate surfaces for global land area, International Journal of Climatology, 37(12), 4302–4315, doi:10.1002/joc.5086, 2017.

Food and Agriculture Organization (FAO). Global Ecological Zones for FAO Forest Reporting: 2010 Update; FAO: Rome, Italy, 2012.

Guevara, M. and Vargas, R.: Downscaling satellite soil moisture using geomorphometry and machine learning, edited by B. Poulter, PLOS ONE, 14(9), e0219639, doi:10.1371/journal.pone.0219639, 2019.

Mascaro, G., Ko, A. and Vivoni, E. R.: Closing the Loop of Satellite Soil Moisture Estimation via Scale Invariance of Hydrologic Simulations, Scientific Reports, 9(1), doi:10.1038/s41598-019-52650-3, 2019.

Møller, A. B., Beucher, A. M., Pouladi, N. and Greve, M. H.: Oblique geographic coordinates as covariates for digital soil mapping, , doi:10.5194/soil-2019-83, 2019.

Wieder, W.R., J. Boehnert, G.B. Bonan, and M. Langseth. 2014. Regridded Harmonized World Soil Database v1.2. Data set. Available on-line [http://daac.ornl.gov] from Oak Ridge National Laboratory Distributed Active Archive Center, Oak Ridge, Tennessee, USA.
http://dx.doi.org/10.3334/ORNLDAAC/1247.

---

## Author Comment (AC2) · 29 Jan 2020

**"Gap-Free Global Annual Soil Moisture: 15km Grids for 1991–2016" by Mario Guevara et al. Responses to anonymous reviewer #2**

REVIEWER COMMENT: OVERVIEW
The study has developed a gap-filled, and downscaled with topography-derived information, long-term annual (15 km) global soil moisture dataset based on the ESA CCI satellite soil moisture products. The assessment of the dataset with respect to in situ observations has been carried out through annual comparisons as well as in terms of long-term trends (1991-2016).

REVIEWER COMMENT: GENERAL COMMENTS
The paper is mostly well written and clear. The topic of the paper is interesting for the readership of Earth System Science Data as a global scale gap-filled annual soil moisture dataset is surely useful for many applications. However, I believe the paper needs major changes before the publication as several parts are not properly described and other sections need to be improved or summarized. I have listed below my comments with the indication of their relevance.

AUTHORS RESPONSE: We appreciate the reviewer comments, the recognition of the importance of this dataset, and support for the possible publication of this manuscript. We have clarified and improved the description of methods and improved the manuscript following these comments and those from other reviewers.

REVIEWER COMMENT:1) MAJOR: The factors used in the downscaling and gap-filling algorithm should be described in details. Two figures (Figures 2 and 3) are not mentioned in the text. It seems a part is missing. The reader needs to know the details on the methodology employed and which factors have been found to be more important. Which Digital Elevation Model is used?

AUTHORS RESPONSE: In a revised version we have included more information about the prediction factors used in our prediction framework and the source elevation data. The source of the DEM is mentioned in the datasets section of the manuscript.

REVIEWER COMMENT: Additionally, other static factors such as vegetation and soil types are not considered. Why?

AUTHORS RESPONSE: Our initial objective was to test the predictive capacity of topographic terrain parameters derived from a single source of information (elevation), considering that a satellite soil moisture pixel is representative of soil moisture of the spatial configuration of climate and ecological conditions (within each pixel) for a specific period of time.
        In the revised version we have included in our prediction framework (as prediction factors) bioclimatic and soil classes static information. We compare the predictive capacity of topographic patterns in relation to these bioclimatic and soil type classes, but we found no significant differences in model performance. Therefore, we conclude that a parsimonious model based on topographic terrain parameters is an alternative approach for downscaling soil moisture while preventing potential spurious correlations (in subsequent analyzes) by adding bioclimatic and soil classes information as prediction factors.

[Figure]

Figure R2_1 Soil moisture across Tropical Rain Forests of the world based on the data available in the ESA-CCI soil moisture product (4.5) for the year 2018 (a). We show the soil moisture prediction (b), the soil moisture prediction variance using only the data available for Tropical Rain Forests (c). Note that the correlation between observed and predicted decreased to 0.62, most likely due to the limited information for modeling these ecosystems, however the root mean squared error is comparable with a model using all global data (e.g., <0.04).

REVIEWER COMMENT: The discussion on the approach employed for downscaling and gap-filling needs to be included in the paper. Why do the authors select such approach?

AUTHORS RESPONSE: We included more information on model selection in the revised version of our manuscript. We used a *kknn* algorithm because it is fast in comparison with other modeling approaches that are computationally more expensive (i.e., deep learning). We do not focus in finding the "best method" for predicting soil moisture, but highlight that the machine (computer-assisted statistical) learning between topographic constraints and satellite soil moisture could benefit the spatial representation of soil moisture grids. Algorithms such as Random Forests or Support Vector Machines are examples of conventional machine learning methods that can also be used as regressors for satellite soil moisture gridded surfaces. We decided to use 'fast and effective' *kknn* to generate a baseline of predictions that can be reproduced in hours (annual 1991-2018) using conventional laptops (i.e., 6GB of RAM) and a flexible framework across multiple users of soil moisture information. We recognize that other forms of statistical learning such as deep learning or ensemble learning could increase the accuracy our predictions. Increasing the accuracy of the *kknn* predictions combining multiple forms of statistical learning (e.g., ensemble learning) could be an emergent objective for future work.

REVIEWER COMMENT: 2) MAJOR: I have found particularly challenging performing gap-filling over dense forest regions in the Amazon and in Congo. Satellite soil moisture data cannot be used in such regions due to dense forest that mask the soil moisture signal. How is it possible to extend the signal there only based on topography?

AUTHORS RESPONSE: There are sparse pixels with soil moisture data across specific areas of the tropical rain forest such as the Amazon or Congo (Figure R2_1a) that are useful for modeling and training soil moisture predictions using our approach (Figure R2_1b). We highlight that our approach is a machine learning algorithm that finds relationships from the provided multivariate space to predict patterns (i.e., spatial gaps). Furthermore, in the revised version we include uncertainty estimates so model outputs could be interpreted based on their spatial uncertainty values.

In the revised version we compare the accuracy of including new prediction factors to account for ecological and climate variability (e.g., presence or absence of bioclimatic features). Soil type information (e.g., harmonized world soil database) was also included to account for the capacity of soil to retain water across these areas with low availability of soil moisture data in the ESA-CCI soil moisture product. Our models were replicated multiple times using different combinations for training and validating models and now we can report a surrogate of model based uncertainty (accounting for the variance of models to multiple data variations) (Figure R2_1c).

REVIEWER COMMENT: I would suggest to perform a more detailed validation in these areas. I strongly suggest to perform a comparison with modelled datasets (e.g., ERA5 soil moisture) to have an assessment of the performance over dense vegetated areas. A similar comment can be done for high latitude areas in which frozen soils and snow completely mask the soil moisture signal. Please perform a detailed validation over these areas, too.

AUTHORS RESPONSE: We have improved our validation exercise across these areas (e.g., tropical forests or high latitude areas) searching for available soil moisture data across the published literature. We find a few sites with available data in a tropical rain forest of southeast Mexico (Vargas et al., 2012) and across tropical forests of Brazil (Saleska, et al., 2013) for a total of 9 new sites across tropical areas (in addition to the original sites available in the ISMN). We found good agreement between our predictions and the ESA-CCI available pixels (only those recognized as pixels of high quality by the ESA-CCI), with field soil moisture estimates (in all cases the correlation between observed and predicted r=>0.8).  In addition, using in situ annual precipitation (Bond-Lamberty and

Thomson, 2018), we report higher correlation between our soil moisture predictions and in-situ precipitation records, compared with the original ESA-CCI (e.g., from r=0.31 to r=0.38 in the tropics and r=0.40 to r=0.51 in temperate areas). We believe that this is good alternative comparison for validating and interpreting soil moisture predictions as previous studies have described the coupling between soil moisture and precipitation across multiples scales of available soil moisture and precipitation estimates (Koster, et al., 2004, McColl et al., 2017).

REVIEWER COMMENT: 3) MAJOR: The trend analysis is very interesting. However, as above, we need more details on how trends are computed. For instance, in situ stations are available only at some points over the Earth, are the same locations used with the satellite-derived datasets? If not, the comparison is wrong.

AUTHORS RESPONSE: We improved the description of the methods in the revised version of our manuscript. We clarify that the comparison was done at the annual scale (i.e., annual means) using only pixels where there was a spatial match with the sites available in the ISMN.

REVIEWER COMMENT: Similarly, in situ stations are not available every year, and for the full year. How are the data aggregated in time and space? These details are needed. I expect the results are strongly impacted to these choices.

AUTHORS RESPONSE: We aggregated all available records of the ESA-CCI in an annual basis and the resulting yearly means were used to train a model for each year (Table 1 of submitted paper shows the number of pixels for each year). We recognize that there is limited soil moisture field information for validating models and satellite soil moisture estimates across large areas of the world. We used all information within each ISMN station aggregated in an annual basis (> 8000 tables containing several gigabytes of soil moisture information) and each data/year was used to validate the soil moisture predictions also in a yearly basis. We argue that the effect of missing data across in situ measurements is diluted when aggregating all available data at the global scale (i.e., calculating a global mean). We clarify that the comparison was done at the annual scale (i.e., annual means) using only pixels where there was a spatial match with the sites available in the ISMN.

REVIEWER COMMENT: 4) MODERATE: The machine learning downscaling approach provides soil moisture data with a resolution higher than the original ESA CCI product. However, I am always doubtful on these downscaling approaches as instead of resolution it should be higher spatial sampling. The higher spatial resolution should be tested, but I am aware it is very hard to do (I have this comment for all downscaling studies).

AUTHORS RESPONSE: We recognize that there is a compromise between where and when to sample across scales. We also recognize that all global studies are limited with the available information of global networks, and local studies (across multiple ecosystems and regions of the world) are needed to better test satellite soil moisture downscaling approaches. We highlight that our main focus is to provide a downscaled soil moisture product that improves the spatial representation of the ESA-CCI

and that is independent of climate- or vegetation-related variables (to avoid potential spurious correlations in further analyzes). That said, the the ESA-CCI satellite soil moisture product showed the lower slightly lower accuracy against field data in the ISMN; thus, supporting the applicability of this approach to downscale satellite-derived soil moisture.  In addition, using in situ annual precipitation (Bond-Lamberty and Thomson, 2018), we report higher correlation between our soil moisture predictions and in-situ precipitation records, compared with the original ESA-CCI (e.g., from r=0.31 to r=0.38 in the tropics and r=0.40 to r=0.51 in temperate areas). We believe that this is good alternative comparison for validating and interpreting soil moisture predictions as previous studies have described the coupling between soil moisture and precipitation across multiples scales of available soil moisture and precipitation estimates (Koster, et al., 2004, McColl et al., 2017).

REVIEWER COMMENT: The authors should demonstrate that the downscaled product is able to reproduce features at higher resolutions with respect to the parent ESA CCI product. It is not done in the paper, that's why I believe higher spatial sampling, and not spatial resolution, is more appropriate.

AUTHORS RESPONSE: We found a larger range of soil moisture predicted values compared with the original ESA-CCI soil moisture product. We found a temporal trend at the places of field stations that is more similar between the field data and our predictions compared with the ESA-CCI soil moisture product. Please note that the main purpose of the model is to reproduce the signal of satellite soil moisture using as reference the relationship that it maintains with topographic data. This is a regression problem were the satellite soil moisture measurements (for a specific time across an area, a pixel under a approximately the same vegetation type or general climate condition) are statistically related to multiple quantitative topography surrogates.

We believe that a spatial resampling (e.g., Figure R2_2a) is just a change of spatial resolution by using simple algorithmic approaches across the orthogonal relationship of the variable itself with the latitude and longitude plane. In contrast, our soil moisture predictions (e.g., Figure R2_2b) are replicated and there is a learning process on each iteration in order to maximize the selection of optimal parameters and maximizing the prediction error given a specific spatial resolution defined by the topographic prediction factors. We understand that this represents a conceptual and semantic debate and we believe that we have improved the description of this empirical modeling approach applied to soil moisture in the new version of our manuscript.

[Figure]

Figure R2_2 Comparison between simple spatial resampling (bilinear) applied to the ESA-CCI product from 27 to 5 km grids (a) and a modeling output using our proposed framework using 5km grids across France (b). We present this example to highlight differences between resampling and prediction using our framework and also the applicability and flexibility across scales. See also Guevara and Vargas 2019.

REVIEWER COMMENT: 5) MODERATE: Several performance scores have been used in the paper. However, I don't think it is necessary to use all of them. The authors should discuss what information each performance score is providing for the assessment of the dataset, not simply to list many numbers. Indeed, Tables 2 and 3 are hard to read and not informative. Please summarize only the more relevant scores in a figure.

AUTHORS RESPONSE: We summarized the description in the accuracy numbers using a quantile plot. We believe that his new figure (Figure R2_3) is useful to visualize and compare differences and similarities between field soil moisture, original ESA-CCI soil moisture and modelled soil moisture. We highlight that in the revised version we include a comparison of the predictive capacity using only terrain parameters and another model including terrain parameters, bioclimatic features and soil type classes. These new results support our conclusion that a parsimonious model only using terrain parameters is a good alternative approach for downscaling satellite-derived soil moisture.

[Figure]

Figure R2_3 Evaluation of soil moisture predictions based on quantiles. The relationship between the ESA-CCI and the ISMN in an annual basis (a). We show the relationship between the ISMN field soil moisture and our predictions based on terrain parameters (b) in relation with a model using bioclimatic
and soil type classes as prediction factors (c). Blue line is a perfect model. Blue histogram is from training data and gray histogram are from model predictions.

REVIEWER COMMENT: 6) MAJOR: The range of values of ESA CCI soil moisture products has
little value, as the satellite products are rescaled to match the range of variability of modelled soil moisture from GLDAS. Therefore, the range of values is that obtained from GLDAS. For the analysis shown in Figure 4, and similarly for the trend analysis, the soil moisture
datasets should be rescaled between the minimum and maximum of each time series and expressed as relative soil moisture (between 0 and 1). Then the data should be aggregated and the range of values and the trends can be assessed.

AUTHORS RESPONSE: We agree with the reviewer and the pixel-wise soil moisture trends detected at the global scale used the downscaled soil moisture predictions are now provided in percentage of
change to avoid issues associated with the dimensions of input data. We also report soil moisture trends in percentage of change comparing gridded and field based soil moisture estimates at the places of field stations in the ISMN.

7) MODERATE: I believe the discussion section must be rewritten. General results
are mostly discussed, whereas it should be closely related to the results shown in the paper. I believe it should be shorter and better focused.

AUTHORS RESPONSE: We have improved the narrative of our discussion section and main findings in the revised version of our manuscript.

REVIEWER COMMENT: SPECIFIC COMMENT (L: line or lines)
L307: Why the "angle between satellite sensors and the earth surface" is useful for
determining soil moisture? It has no sense and I believe it is wrong.

AUTHORS RESPONSE: We meant to say that topography affect the distance between the satellite and the earth surface; therefore, it could be correlated with the satellite soil moisture signal (which is a hypothesis supported with the data analyzed in this study).

REVIEWER COMMENT: RECOMMENDATION
Based on the above comments, I suggest a major revision before the possible publication on Earth System Science Data.

AUTHORS RESPONSE: We appreciate the comments of the reviewer as they have been very useful to improve the overall revised manuscript.

References:

Bond-Lamberty, B.P., and A.M. Thomson. 2018. A Global Database of Soil Respiration Data, Version 4.0. ORNL DAAC, Oak Ridge, Tennessee, USA. https://doi.org/10.3334/ORNLDAAC/1578

Guevara, M. and Vargas, R.: Downscaling satellite soil moisture using geomorphometry and machine
learning, edited by B. Poulter, PLOS ONE, 14(9), e0219639, doi:10.1371/journal.pone.0219639, 2019.

Koster, R. D.: Regions of Strong Coupling Between Soil Moisture and Precipitation, Science,
305(5687), 1138–1140, doi:10.1126/science.1100217, 2004.

McColl, K. A., Alemohammad, S. H., Akbar, R., Konings, A. G., Yueh, S. and Entekhabi, D.: The global distribution and dynamics of surface soil moisture, Nature Geoscience, 10(2), 100–104,
doi:10.1038/ngeo2868, 2017.

---

## Author Comment (AC3) · 29 Jan 2020

**"Gap-Free Global Annual Soil Moisture: 15km Grids for 1991–2016" by Mario Guevara et al. Responses to anonymous reviewer #3**

REVIEWER COMMENT: General comments: While seeking a higher resolution global soil moisture product is certainly a laudable goal, I find the methods used in this paper lack a credible connection between ground measurements and the remotely sensed data. The authors use machine learning to generate regressions between multiple topographic variables thought to be related to soil moisture that are available at higher resolution with coarse resolution satellite data. It is difficult for the reader to discern if actual new information results from the downscaling because there is not a clear connection made between insitu soil moisture measurements, their physical connection to the chosen topographic variables used, and resultant satellite measurements.

AUTHOR RESPONSE: In a revised version we have improved the introduction and methods sections in a revised version to highlight why topographic terrain parameters are hydrologically meaningful. We would like to clarify that in situ observations are used for validating purposes only and were not used for developing (only testing) our downscaled soil moisture product. In this study, the main purpose was to compare if the fusion of satellite soil moisture records and elevation-derived terrain parameters (by the means of an empirical modeling approach, not physical) was useful to fill gaps of satellite estimates. The physical connection between soil moisture and terrain attributes is that these attributes regulate the overland flow and the potential solar radiation income (REF). Both processes are controlled by topography and therefore should show a direct influence on soil moisture patterns. We have also included more information supporting the use of these hydrologically meaningful terrain parameters for predicting soil moisture patterns, an emergent research opportunity on Geomorphometry. Finally, we clarified that our approach is not a process-based model, it is an empirical approach using machine learning and taking advantage of the large multivariate space of topographic parameters across the world (trained using ESA-CCI soil moisture) to predict soil moisture at 15 km resolution across the world. We highlight that this approach results in an improvement of the spatial resolution of soil moisture across the world and a better match with in situ soil moisture information (from the ISMN) when compared with the original ESA-CCI soil moisture product.

REVIEWER COMMENT: To do so, requires first demonstrating the rigor of the methodology over a much smaller and better measured area, such as the area of the International Soil Moisture Network.

AUTHOR RESPONSE: We fully agree with this comment. Our approach has already been tested across the contiguous United States (Guevara and Vargas, 2019). Thus, this study is an extension of that demonstrated methodology and now is applied to the global scale.

REVIEWER COMMENT: Extending the algorithm to areas that are hydrogeomorphically and climatically distinct from the existing soil moisture measurement networks cannot be result in credible data. I do not recommend publication of this work.

AUTHOR RESPONSE: We respectfully believe this is a misunderstanding from the reviewer. We clarified that or modeling approach does not use data from the existing soil moisture measurement networks to predict soil moisture across the world. Data from the ISMN is only used for validation purposes and was never used for training the model.

As explained in previous work (i.e., Gessler et al., 2009, Florinsky, 2012, MacMillan et al., 51 2016), terrain parameters (such as those we used for predicting soil moisture) both influence the accumulation of surface geological materials and reflect this spatial distribution. Terrain parameters such as the wetness index or the relative slope position both influence the flow and accumulation of 54 moisture and reflect it. Terrain parameters both influence the spatial patterns of distribution of vegetation/land use and reflect these patterns. Our methods is based on geomorphometry which differs from a purely physical hydrological model (i.e., process-based models). That said the novelty of our 57 study is that it takes the ESA-CCI and the terrain parameters in to a machine learning approach and finds relationships within the multivariate statistical space. The training data is the satellite soil moisture pixels in the latest version of the ESA-CCI, which are soil moisture values representative of a 60 mosaic of ecological and environmental conditions (for a specific time) within an area (pixel) of around 27km of spatial resolution. The strength of a machine learning model as *kknn* is to find non-linear relationships in a complex multivariate space to predict patterns.

These patterns are dependent of training data (i.e., ESA-CCI soil moisture) and there are pixels with valid soil moisture data across dense vegetation areas or high latitudes that are used to train our model for soil moisture predictions (Fig_R3_1). We re-analyzed our datasets and updated soil moisture 66 predictions including multiple sources of prediction factors (e.g., bioclimatic, soil type information) in order to test the reliability of our prediction framework.

The revised version of our manuscript included new data from other published soil moisture 69 values to expand our validation dataset (Vargas et al., 2012, Saleska, et al., 2013). ). In addition, using in situ annual precipitation (Bond-Lamberty and Thomson, 2018), we report higher correlation between our soil moisture predictions and in-situ precipitation records, compared with the original ESA-CCI 72 (e.g., from r=0.31 to r=0.38 in the tropics and r=0.40 to r=0.51 in temperate areas). We believe that this is good alternative comparison for validating and interpreting soil moisture predictions as previous studies have described the coupling between soil moisture and precipitation across multiples scales of 75 available soil moisture and precipitation estimates (Koster, et al., 2004, McColl et al., 2017).

REVIEWER COMMENT: Major comment: Much more information is needed about the regression methods and parameter selection.

AUTHOR RESPONSE: In the revised version we improved the narrative of the kernel based machine learning algorithm. The main parameters of this method (*kknn*) are the kernel type (used to convert distances between neighbors points in the statistical space, to weights) and the k parameter (the number 84 of neighbors used to calculate a weighted average in regression). These parameters are selected in our modeling approach automatically using repeated cross validation (Table 1 of submitted paper). The terrain parameters derived by the means of geomorphometry from the digital elevation model are 87 another component of our modeling framework, used prediction factors (soil moisture covariates) in the *kknn* approach.

REVIEWER COMMENT: It would be helpful to discuss what each parameter is mathematically and why it can be useful in a soil moisture prediction context.

AUTHOR RESPONSE: We explain with more detail (in our revised manuscript) why the terrain parameters derived from elevation data using Geomorphometry are hydrologically meaningful. 96 Detailed information on these terrain parameters is reported in our previous study across the conterminous United States (Guevara and Vargas 2019; see table: https://doi.org/10.1371/journal.pone.0219639.s007).

REVIEWER COMMENT: Can it be demonstrated over smaller areas that there is a valid argument for using these variables, some or all of them.

AUTHOR RESPONSE: Across the conterminous USA, we found an increase of nearly 25% (when compared with the original ESA-CCI soil moisture product) of accuracy validating our predictions against the North American Soil Moisture Database (https://doi.org/10.1371/journal.pone.0219639.g005).

REVIEWER COMMENT: Explain in detail the cross validation process used and why.

AUTHOR RESPONSE: Cross validation is a family of common re-sampling techniques that are used to analyze the sensitivity of models (in this case) to variations of available data for training purposes. Multiple models are generated using multiple proportions of available data and validating with subsets of information that are leaved out of these models. This process is repeated several times until capturing the magnitude of variance associated with the models based on the data subsets. In the revised version we included more information and references about cross validation strategies for re- sampling and bootstrapping prediction models.

REVIEWER COMMENT: It is unclear if there is a single model developed and applied to all years or separate models for each year. Are all topographic parameters used in the model in all years? What is their weighting?

AUTHOR RESPONSE: We generated and cross validate a model for each year of available data. Table 1 (in the original manuscript) shows the results of cross validation and available data for each model/year. All topographic parameters are used on each model/year. During the cross validation, kknn uses a kernel form to convert distances in weights that are used for prediction. These distances are different from one place to another as the pattern recognition includes all the points and their k neighbors. Using the independent residuals of each model realization during the cross validation strategy, the optimal weights are selected using as indicators the root mean squared error and the correlation between observer and predicted (also included in Table 1 of the original manuscript). After all model realizations including all possible kernel combinations (e..g, "rectangular", "triangular", "epanechnikov","gaussian", "rank", "optimal") and increasing the number of neighbors (k) systematically (e.g., from 2 to 25, See Table 1) we find for each year the optimal weights maximizing the correlation between observed and predicted and minimizing the root mean squared error. In our reanalysis, we have included a novel variable importance analysis (by permutation) for kknn, that allows to identify which are the most important variables in the overall prediction using this kernel based algorithm.  We have clarified these statements in the revised version of the manuscript.

REVIEWER COMMENT: Major comment: Extending the results to areas without soil moisture measurements
(whether in situ or remotely sensed values) without any validation is not an improvement over current spatial data. It is highly suspect and could lead to inappropriate
applications using what is in effect non-data.

AUTHOR RESPONSE: We agree that any modeling output should be interpreted and used carefully as it is not a direct measurement. However, we performed a robust cross validation strategy including data available in the ESA-CCI representing all environmental conditions and provide information about
data-model agreement. Then we validate with field data only at the places (pixels) containing in situ information in the International Soil Moisture Network and additional sites in tropical regions (9 more sites in the revised version of the manuscript). We argue that there are limited but representative
satellite soil moisture records across these areas that can be used to train models for global soil moisture patterns (Fig_R3_1).

In the revised version we include more datasets (Fig_R3_1c) and more validation information to
enrich the discussion about the reliability of our prediction framework for soil moisture (e.g., local precipitation measurements). We highlight that we are following a conceptual and data-driven framework assuming that comparing and testing multiple modeling approaches is still required to better
understand the implications of soil moisture on ecological patterns across areas where no soil moisture information is available. Thus, our results provide evidence that our soil moisture product is useful to predict soil moisture patterns in places with low availability of satellite soil moisture estimates.
Finally, in the revised version we now include uncertainty estimates to complement this study and inform users and applications about model performance across the world (Fig_R3_2).

[Figure]

Fig_R3_1 Available information in the ESA-CCI for the year 2018 in the latest version product
(4.5). We show the available information across Polar environments (a) and across dense
vegetated areas and tropical rain forests (b). We also show a recent database with in situ
information of climate variables and ecological properties that we used to support the
reliability of our prediction framework in a revised version of our paper (c).

[Figure]

Fig_R3_2 Mean soil moisture for the year 2018 from the ESA-CCI (a). Spatial prediction of soil moisture for the year 2018 across 15km grids (b) and prediction variances based on bootstrapping the spatial prediction model.

REVIEWER COMMENT: Moderate comment: I found the paper very repetitive and in need of detailed editing.

AUTHOR RESPONSE: We have revised the narrative and overall organization in the revised manuscript.

References

Bond-Lamberty, B.P., and A.M. Thomson. A Global Database of Soil Respiration Data, Version 4.0.
ORNL DAAC, Oak Ridge, Tennessee, USA. https://doi.org/10.3334/ORNLDAAC/1578, 2018

Florinsky, I. V.: Influence of Topography on Soil Properties, Digital Terrain Analysis in Soil Science
and Geology, 145–149, doi:10.1016/b978-0-12-385036-2.00008-0, 2012.
Gessler, P., et al.,  Chapter 28 The Future of Geomorphometry, Developments in Soil Science, 637–
652, doi:10.1016/s0166-2481(08)00028-7, 2009.
Guevara, M. and Vargas, R.: Downscaling satellite soil moisture using geomorphometry and machine
learning, edited by B. Poulter, PLOS ONE, 14(9), e0219639, doi:10.1371/journal.pone.0219639, 2019.
Koster, R. D.: Regions of Strong Coupling Between Soil Moisture and Precipitation, Science,
305(5687), 1138–1140, doi:10.1126/science.1100217, 2004.

MacMillan et al., A hydrologically explicit, spatially exact, classification of landforms for Canada at
1:500,000 scale., EGUGA, EPSC2016-13103
https://ui.adsabs.harvard.edu/abs/2016EGUGA..1813103M/abstract (Accessed 23 December 2019),
2016.

McColl, K. A., Alemohammad, S. H., Akbar, R., Konings, A. G., Yueh, S. and Entekhabi, D.: The
global distribution and dynamics of surface soil moisture, Nature Geoscience, 10(2), 100–104,
doi:10.1038/ngeo2868, 2017.

Saleska, S.R., H.R. da Rocha, A.R. Huete, A.D. Nobre, P. Artaxo, and Y.E. Shimabukuro. 2013. LBA-
ECO CD-32 Flux Tower Network Data Compilation, Brazilian Amazon: 1999-2006. Data set.
Available on-line [http://daac.ornl.gov] from Oak Ridge National Laboratory Distributed Active
Archive Center, Oak Ridge, Tennessee, USA http://dx.doi.org/10.3334/ORNLDAAC/1174

Vargas, R.: How a hurricane disturbance influences extreme CO2fluxes and variance in a tropical
forest, Environmental Research Letters, 7(3), 035704, doi:10.1088/1748-9326/7/3/035704, 2012.